# Asymmetric vibrations in the organ of Corti by outer hair cells measured from excised gerbil cochlea
Wei-Ching Lin[1], Anes Macić[1], Jonathan Becker[1] & Jong-Hoon Nam [1,2,3] ✉

Pending questions regarding cochlear amplification and tuning are hinged upon the organ of Corti (OoC) active mechanics: how outer hair cells modulate OoC vibrations. Our knowledge regarding OoC mechanics has advanced over the past decade thanks to the application of tomographic vibrometry. However, recent data from live cochlea experiments often led to diverging interpretations due to complicated interaction between passive and active responses, lack of image resolution in vibrometry, and ambiguous measurement angles. We present motion measurements and analyses of the OoC sub-components at the close-to-true cross-section, measured from acutely excised gerbil cochleae. Specifically, we focused on the vibrating patterns of the reticular lamina, the outer pillar cell, and the basilar membrane because they form a structural frame encasing active outer hair cells. For passive transmission, the OoC frame serves as a rigid truss. In contrast, motile outer hair cells exploit their frame structures to deflect the upper compartment of the OoC while minimally disturbing its bottom side (basilar membrane). Such asymmetric OoC vibrations due to outer hair cell motility explain how recent observations deviate from the classical cochlear amplification theory.

Highly organized truss-like structures characterize the mammalian auditory epithelium (the organ of Corti, OoC). The OoC consisting of the sensory hair cells and their supporting cells, sits on the basilar membrane. Some of the supporting cells, such as the pillar cells, reticular lamina, and Deiters cells together with the basilar membrane, form a mechanically significant frame housing the hair cells. This OoC frame underlies the passive force transmission to the hair cells and active force transmission from the hair cells[1,2].

The OoC is stiff enough to deliver basilar membrane vibrations to the hair cells with a negligible delay. Meanwhile, the OoC is flexible enough to get deformed by motile outer hair cells[3–6]. The cytoskeletal organization of the supporting cells in the OoC suggests substantial stiffness. For example, the pillar cells are packed with microtubules bundled by crosslinking actin fibers[7]. The Deiters cells have bundled microtubules in their soma and process[8]. The reticular lamina is like mosaic tiles consisting of the tops of the pillar cells, the cuticular plate of hair cells, and the process of Deiters cells, all abundant in microtubules and actin fibers[9]. The OoC frame, consisting of these stiff components, is far from rigid. The joint between the outer hair cell and Deiters cell vibrates more than other parts of the OoC when outer hair cells are motile[3,10,11]. The Deiters cell is as compliant as the outer hair cell[5]. The reticular lamina bends and stretches over its span[12]. The estimated buckling force of the outer pillar cell is comparable to the force generated by

an outer hair cell (on the order of $10^{-9}$ N[8]). The deformable OoC must have implications for active force transmissions.

Tomographic vibrometry has revealed intriguing interactions between the deformable OoC and the basilar membrane that vary depending on stimulating frequency and level, suggesting distinct routes of active and passive force transmissions in the OoC. For instance, the vibrations in the upper lateral part of OoC show amplifications well below the characteristic frequency (CF), unlike the basilar membrane[13,14]. The locus of the largest vibration shifts from the basilar membrane to the upper part of the OoC as the sound pressure level decreases[3]. The deflection of the hair cell bundle relative to the basilar membrane depends on frequency[15]. Recent observations, including these studies, are incongruent with the classical cochlear mechanics, where basilar membrane vibrations represent cochlear amplification[16]. To better appreciate recent findings of tomographic vibrometry, active and passive force transmission within the OoC frame must be characterized unequivocally. Despite this need, specifying the force transmission in the OoC from vibration measurements has been challenging for the following reasons. First, in live cochlear experiments, it is tricky to isolate active OoC motion due to outer hair cell motility from passive motion due to acoustic pressures. Second, it is difficult to resolve the motions of OoC sub-components unless imaged through the round window

[1]Department of Mechanical Engineering, University of Rochester, Rochester, NY, USA. [2]Department of Biomedical Engineering, University of Rochester, Rochester, NY, USA. [3]Department of Neuroscience, University of Rochester, Rochester, NY, USA. ✉e-mail: jong-hoon.nam@rochester.edu

or a surgical opening[12,17]. Third, it is difficult to choose or determine the incidence angle of vibrometry[11,18].

In this study, we characterized active and passive force transmission in the OoC by analyzing the 2-D vibrations of excised gerbil cochleae. Our approach circumvented the difficulties in existing studies. We could isolate the active and passive vibrations of OoC. By removing the bones that obstruct the optical path, the vibrometry image became fine enough to resolve individual cells confidently, such as a micron-thick column of the outer pillar cell. We reduced ambiguity in measuring radial and transverse motions by using a wider range of incidence angles (> 30 degrees) at the close-to-true radial cross-section (< 5 degrees of longitudinal tilt). We focused on 2-D (radial and transverse) motions of the basilar membrane, reticular lamina, and outer pillar cell because they construct the frame for active and passive force transmission in the OoC. Our results provide a coherent explanation of how outer hair cells exploit their structural frame to enhance vibrations of the upper part of the OoC relative to those of the basilar membrane.

## Results

All results of this study are regarding 2-D vibrations in a radial section of the gerbil cochlea. OoC vibrations were measured at two orientation angles (Fig. 1, Fig. 2a, c) and decomposed into radial and transverse motion

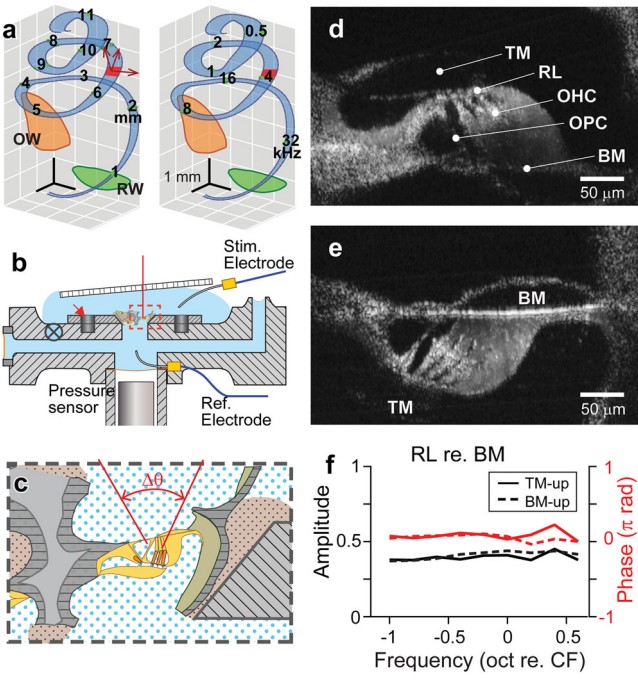

**Fig. 1 | Excised cochlear preparation. a** The measurement location of this study is indicated with the red patch. It is 6.5 mm from the basal end with an expected characteristic frequency of 4 kHz. The blue coils represent the basilar membrane. The numbers on the left and right coils indicate the distance in mm according to Plassman et al.[41]. and the CF in kHz according to Müller[40]. The orange and green areas indicate the oval (OW) and round (RW) windows. **b** Excised cochlear turn was placed in a plastic disk. The disk was placed securely on a custom-fabricated microfluidic chamber using magnets (red arrow). The tissue was subjected to either mechanical agitations through an elastic window on the left of this cartoon or electrical agitations through a pair of electrodes. **c** An enlarged view of the micro-chamber slit where the targeted OoC was located. The red vertical lines represent the range of laser beam directions of the optical coherence tomography (OCT). The dull pink region with dots indicates the glue to adhere and seal the preparation onto the magnet disk. **d, e** The cochlear tissue was placed so that the tectorial membrane or the basilar membrane faced upward (called the 'TM-up' or 'BM-up' preparation). TM: tectorial membrane. RL: reticular lamina. OHC: outer hair cell. DC: Deiters cell. BM: basilar membrane. **f** The reticular lamina vibrations with respect to the basilar membrane. The orientation of tissue (TM-up or BM-up) did not affect vibration responses. Data set used: M1113 of 2022, M0126 of year 2023.

components before being reconstructed into 2-D motion trajectories (Fig. 2e, f, k, l). Responses to mechanical and electrical stimulation are presented together in each figure, hereafter, referred to as 'M-stim' and 'E-stim' responses. Followed by the overall OoC complex vibration analysis due to M-stim and E-stim in Fig. 2, the responses of three elements that form the frame of OoC (basilar membrane, reticular lamina, and outer pillar cell) are analyzed in Figs. 3–6. Relevant animations (Supplementary Movies 1–6) accompany the figures to assist readers' appreciation.

In studies with live animals, active and passive cochlear responses often refer to sensitive and insensitive cochlear responses. We used excised (ex vivo) cochleae to study the vibrations due to outer hair cell motility (E-stim) and fluid pressures (M-stim). In this study, we call the vibrations due to E-stim as active responses, although our in vitro preparation is not sensitive.

### Two-dimensional vibration patterns depending on stimulus types

Responses to M-stim and E-stim showed characteristics of insensitive and sensitive cochlear responses of the intact (live) cochlea, respectively (Fig. 2, Supplementary Movie 1). The OoC vibrated nearly in-phase for M-stim (Fig. 2b, d, Supplementary Fig. 1i). In contrast, the vibrating pattern was non-monotonic for E-stim (Fig. 2h, j). M-stim vibrations peaked near the middle of the OoC radial span (Fig. 2e, f), while vibrations due to E-stim were largest in the upper lateral part of the OoC (Fig. 2k, l). For M-stim, the tectorial membrane and the OoC vibrated as if structural bodies rotate roughly about the bony edges (red asterisks in Fig. 2e, f). In contrast, the center of rotation was not apparent in E-stim responses (Fig. 2k, l). The motion trajectories were transversely dominant in M-stim responses, while E-stim responses showed greater radial motions, especially around outer hair cells. (Fig. 2e, f versus Fig. 2k, l). For M-stim, vibration amplitudes of the reticular lamina and the basilar membrane were comparable. In contrast, the basilar membrane vibrated much less than the reticular lamina in E-stim. This feature of relative vibration amplitude between the top and bottom of OoC will be analyzed in further detail later in this paper. Overall, these differences between M-stim and E-stim responses are consistent with passive and active responses in experiments with intact cochleae, respectively[3,15]. Our observations corroborate the notion that the characteristics of sensitive cochlear responses originate from outer hair cell motility.

### Two premises regarding basilar membrane vibration pattern over its radial span

The basilar membrane is considered to vibrate transversely at its primary mode. Note that this brief statement contains two premises that have been taken without rigorous discussion. One is that the basilar membrane vibrates at its 'primary mode' (i.e., a half-sine or raised-cosine that has a peak in the middle of its span). For this assumption of primary mode, there are supporting observations. Over its radial span, the basilar membrane deformed as if the arcuate and pectinate zones were a pair of saloon doors[19]. Our previous studies[5,20] showed a similar saloon-doors pattern of vibration, but the bandwidth of hydrodynamic deformation was narrower than the deformation due to hydrostatic pressure, reminiscent of vibration mode transition.

The other premise is that the basilar membrane motion is 'transverse'. Most measurements of cochlear mechanics have been one-dimensional. The axis of measurements was considered to represent the transverse motion of the basilar membrane, implying that the transverse motion is of primary interest and that the radial motion is negligible. Notably, a 2-D vibration measurement showed that the basilar membrane's motion in the radial direction was non-negligible as compared to its transverse motion[15].

We examined the two premises regarding basilar membrane motion. In Fig. 3, decomposed radial and transverse vibration patterns of the basilar membrane were presented for the M-stim and E-stim (also see Supplementary Movie 2). For M-stim, consistent with the previous studies[5,19,20], the peak displacement occurred near the joint between the arcuate and pectinate

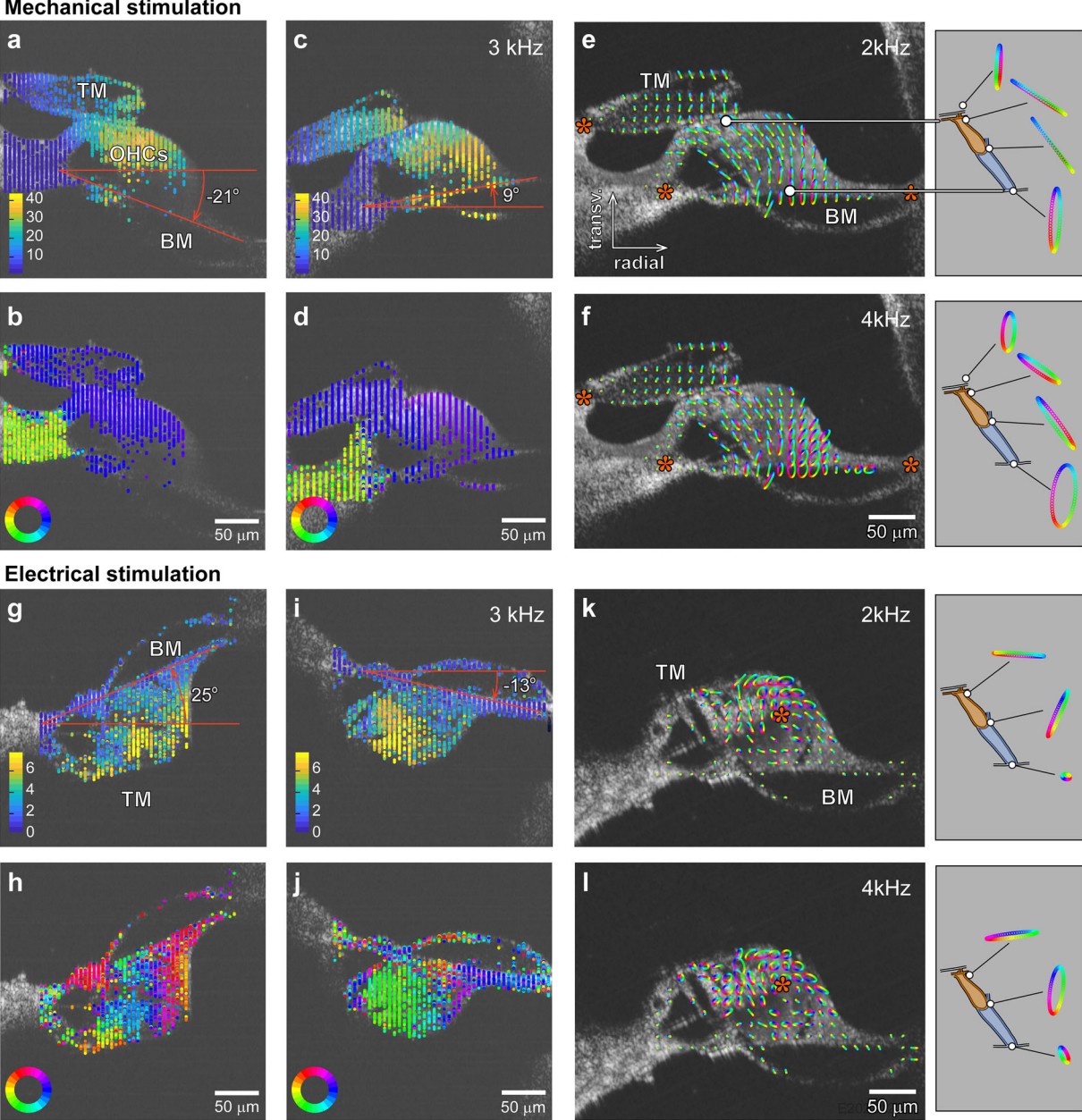

**Fig. 2 | Two-dimensional vibration measurement.** Two sets of OoC vibration measurement: one set due to mechanical stimulation with the 'TM-up' orientation (**a**–**f**), and the other set due to electrical stimulation with 'BM-up' orientation (**g**–**l**). **a**, **b** Amplitude of vibrations measured at two orientation angles. **c**, **d** Phase of vibrations. A 4-kHz CF location was mechanically stimulated at 3 kHz. **e**, **f** Two-D vibrating patterns when stimulated at 2 kHz and 4 kHz. Each ellipsoid represents a vibration trajectory. The colors in the trajectory correspond to the phase of oscillatory motion. The right sub-panels show the motion trajectory of the second-row outer hair cell and Deiters cell. **g**–**j** Vibrations measured at two orientation angles when the OoC was subjected to trans-epithelial current alternating at 3 kHz. **k**, **l** Two-D motion trajectories of the OoC for 2 and 4 kHz electrical stimulation. The linear scale bars in (**a**, **c**, **g**), and i are in nanometers (nm). The colors of angular scale bars in (**b**, **d**, **h**, **j**) apply to the cyclic trajectories in (**e**, **f**, **k**, **l**) as well. Data sets used: M0928 and E1220 of year 2022. Animated figure: Supplementary Movie 1 (https://doi.org/10.6084/m9.figshare.23710605).

zones ('B' in Fig. 3a). For the transverse motion component of M-stim, the phase at the edge was different from the rest of the OoC, and the difference enlarged, both in value and span, as the stimulating frequency increased (green to light blue colors in Fig. 3b). For the radial motion component of M-stim, we did not find a clear pattern (Fig. 3c, d). The motion trajectories were linear rather than elliptical (Fig. 3f). The transverse motion component was much greater than the radial component (Fig. 3e–h). For E-stim, the basilar membrane vibrating pattern differed from that of M-stim in several aspects. Transverse and radial motion amplitudes and phases are presented in Fig. 3i–l, and motion trajectories in Fig. 3m–p. First, there were multiple peaks in a vibrating pattern, like a higher-order vibration mode (Fig. 3m, o, p). Second, the motion trajectories were elliptical rather than linear (Fig. 3n). Third, the radial motion was as large as the transverse motion (Fig. 3n, o, p). These results in Fig. 3 were the responses at 2.8 kHz at 4 kHz CF location. We observed frequency-dependence in basilar membrane vibration patterns, presented in Fig. 4 (and Supplementary Movie 3). Note that, when presenting the radial and transverse motions they were normalized so that the ratio between the two motion components can be obtained from the presented plots (i.e., normalized by the greater between the radial and the transverse components).

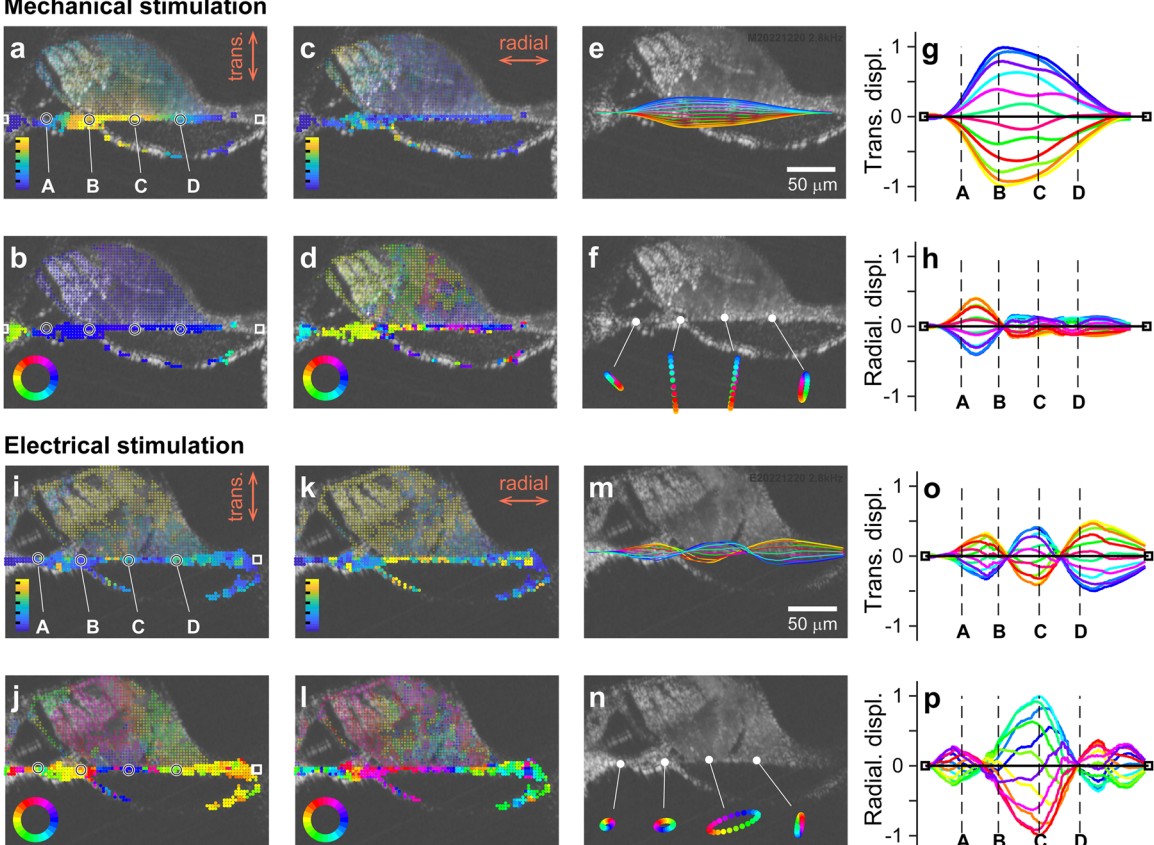

**Fig. 3 | Basilar membrane's vibrating pattern.** A BM-up preparation at a 4 kHz CF location was stimulated mechanically and electrically at 2.8 kHz at two orientation angles. Measured motions were decomposed into transverse (vertical) and radial (horizontal) components. **a**, **b** Transverse vibration due to mechanical stimulation in amplitude and phase. The points of interest represent the middle of the arcuate zone (A), the joint between arcuate and pectinate zones (B), the root of the second-row Deiters cell (C), and the middle of the pectinate zone (D). **c**, **d** Radial vibration due to mechanical stimulation in amplitude and phase. **e** Basilar membrane vibrating pattern. The color of each curve corresponds to the phase of sinusoidal oscillations. **f** Basilar membrane vibration trajectory at four different sites. **g** Transverse vibration over the span. **h** Radial vibration over the span. **i–p** Basilar membrane vibrations to electrical stimulations, presented similarly to (**a–h**). Data sets used: M1220 and E1220 of year 2022. Animated figure: Supplementary Movie 2 (https://doi.org/10. 6084/m9.figshare.23710587).

## Non-negligible radial displacement of the basilar membrane

For M-stim responses, the vibrations remained transversely dominant (i.e., the response amplitudes were larger in Fig. 4a than in Fig. 4b). For E-stim, the vibration amplitude was comparable between the radial and transverse motion, and the motion trajectory was elliptical rather than linear. In Fig. 4c, d, the curves represent the snapshots of transverse and radial vibrating patterns when the vibration peaked. The ratio between transverse and radial amplitudes at their respective peaks was presented for the frequency range between 1 and 8 kHz (Fig. 4e). For M-stim, the basilar membrane motion was transversely dominant below the CF, but the radial component became comparable to the transverse component at frequencies higher than the CF. For E-stim, the motion amplitude was comparable between the transverse and radial components regardless of frequency. Figure 4f presents similar results to Fig. 4e but from four different cochleae. The results were averaged per one-octave bins (the circle and square symbols with error bars). The trend of the transverse versus radial motion ratio persisted across different preparations. A similar trend was observed in live cochlear measurements: The fraction of the basilar membrane's radial motion decreased as the sound pressure level increased or the contribution of outer hair cell motility decreased[15]. Note that the peak shown in Fig. 4e plot is irrelevant to the best-responding frequency. A peak in mechanical gain (m/Pa) versus frequency curve defines the CF. The frequency-dependent curve of the M-stim response in Fig. 4e represents a change in the vibrating pattern (eigenvector or mode).

The results in Fig. 4d should be interpreted as the change in vibrating patterns over frequency for the M-stim. Unlike the case of M-stim, the pattern remained similar despite varying frequencies in the case of E-stim.

## Twist deformation of the basilar membrane due to outer hair cell force

For M-stim responses, the vibration pattern remained similar below the CF (<3 kHz), showing the saloon door pattern. However, as the stimulating frequency approached CF (4 kHz), the indication of a higher-order mode appeared (yellow-black curve in Fig. 4a). The higher-order modes became more evident when the stimulating frequency was well above the CF (>5 kHz, violet curve in Fig. 4a). For E-stim responses, the vibration pattern was like the primary mode at two octaves below the CF (at 1 kHz, blue curve in Fig. 4c). But at frequencies near or above the CF (>2 kHz), the vibrating pattern became more complex than the simple saloon door pattern.

Previous studies reported the radial vibration pattern of the basilar membrane. Some[21,22] reported complex basilar membrane motion similar to our E-stim responses. Meanwhile, others[19,23] observed simple motions like our M-stim responses. Given our results, such previous observations are not necessarily mutually exclusive. In the live cochlea, responses of M-stim and E-stim must coexist. Either response pattern can prevail depending on the combination of active and passive vibration patterns. The contribution of outer hair cell motility depends on the sound pressure level and frequency. The E-stim vibration pattern will prevail at the stimulus level and frequency where cochlear mechanics are nonlinear. Our results suggest that when the

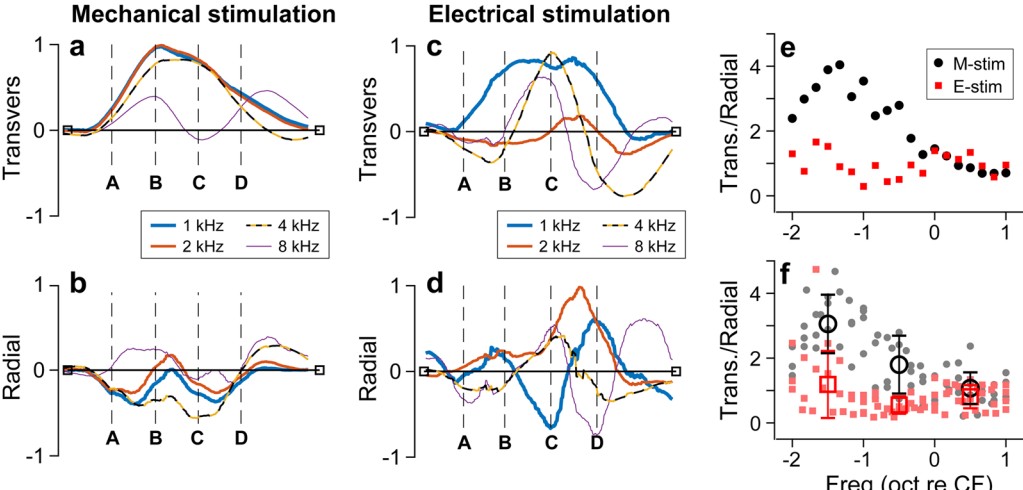

**Fig. 4 | Basilar membrane vibrating patterns at different frequencies.** The locations A–D correspond to those in Fig. 3A. **a** Transverse vibration pattern at four different frequencies of mechanical stimulation. A 4 kHz CF location was mechanically stimulated at frequencies between 1 and 8 kHz. **b** Radial vibration patterns at different frequencies of mechanical stimulation. **c, d** Transverse and radial vibration patterns at different frequencies of electrical stimulation. **e** The ratio between transverse and radial motion versus stimulating frequency. **f** The transverse-radial motion ratio from four measurements for mechanical and electrical stimulations, respectively. Data sets used: M0928, M1212, M1220a, M1220b, E0621, E1212, E1220 from year 2022. Animated figure: Supplementary Movie 3 (https://doi.org/10.6084/m9.figshare.23710611).

stimulating frequency is much lower than the CF, the basilar membrane will vibrate with its primary mode regardless of cochlear nonlinearity (outer hair cell motility) (Fig. 4c). Conversely, when the stimulating frequency is higher than the CF, the basilar membrane will vibrate with its higher-order mode despite the sound pressure level (Fig. 4a). Therefore, it is not surprising that previous studies showed the spectrum between primary and higher-order-mode-like vibrations.

Despite its complex pattern, the basilar membrane deforming pattern due to E-stim is not a higher-order mode. The deforming pattern is caused by the way the outer hair cell force applies to the basilar membrane. Because of the OoC frame mechanics, active outer hair cells twist the basilar membrane (i.e., push-and-pull rather than push-or-pull). We will discuss this point further toward the end of this paper. Superposing the basilar membrane's passive and active vibration patterns suggests an interesting scenario. Outer hair cell motility could facilitate the higher-order mode of passive vibrations.

### The reticular lamina bends due to outer hair cell force
The reticular lamina motion showed distinct patterns depending on the stimulus type (Fig. 5, Supplementary Movie 4). For M-stim, the reticular lamina moved like a rigid bar rotating about the root of the inner pillar cell (Fig. 5a, b). For E-stim, the reticular lamina deflected substantially, inflecting between the second and third-row outer hair cells (Fig. 5h, i). Although the deflecting shape was reminiscent of higher-order mode vibrations, it was not, considering the minimal change in vibrating patterns over the tested frequency range (Fig. 5e, f, l, and m).

To analyze rigid-body versus bending motion depending on the stimulus type, we decomposed the overall motion of the reticular lamina into rigid-body motion and deformation (Fig. 5g, n). The deformation was obtained by subtracting the translational and rotational rigid-body motion from the overall motion (i.e., the top plot of Fig. 5g equals the sum of the middle and bottom plots of Fig. 5g). Figure 5g, h show that the reticular lamina moved like a rigid body by M-stim, but it was bent by E-stim.

Reticular lamina motion has drawn increasing attention recently[12,17,24]. Our results demonstrate the significance of presenting the accurate locus and the orientation of measurement, especially in sensitive cochleae, considering the complex vibration pattern with a small characteristic length. For instance, if one measure reticular lamina motions at the first or third-row outer hair cell (points 'A' or 'D', Fig. 5h–n), the apparent phase of motion could differ by as large as 180 degrees. The phase can be ambiguous depending on the measurement site and angle. Our results suggest that the precise measurement site and view angle should also be presented for a reticular laminar motion measurement to be fully informative.

### The outer pillar cell also bends due to outer hair cell force
Similar to the reticular lamina analysis, outer pillar cell motions were decomposed into rigid body motion and deformation (Fig. 6, Supplementary Movie 5). Due to their truss-like triangular configuration and their core reinforced with microtubules, it seems reasonable to consider the pillar cells as the rigid scaffold of the OoC. This view of the tunnel of Corti as a rigid scaffold is consistent with our M-stim results (Fig. 2e, f and Fig. 6b). When mechanically stimulated, the outer pillar cell rotated about the medial edge of the OoC as if it is a rigid body. In contrast, our E-stim responses are inconsistent with the rigid-body model. When electrically stimulated, the outer pillar cell bent rather than moving like a rigid body (Fig. 6d–f). Our results indicate that the outer pillar cell deflects due to outer hair cell force. The overall pattern of outer pillar cell motion was comparable to that of the reticular lamina (Fig. 6g).

For M-stim, the vibration amplitude was similar at different points along the length of the outer pillar cell (Fig. 6b). For E-stim, in contrast, the motion was largest in the middle, and the moving direction was approximately normal to its length axis (Fig. 6e). The bending deformation suggests that the head and foot of the outer pillar cell are clamped rather than hinged. Out of three data sets of the E-stim case, in two sets, the mid-section bent toward the tunnel of Corti due to contracting outer hair cells like Fig. 6f, but the third set showed the opposite bending direction. Despite this variance, one characteristic was unmistakably consistent. While M-stim resulted in a similar level of vibration amplitude at the head and foot of the outer pillar cell, E-stim resulted in minimal motion at the lower end (the basilar membrane side). We will further discuss the implication of outer pillar cell motion and deformation using Fig. 7.

### Discussion
We characterized the active and passive vibrations of the OoC by decomposing the motions of the outer pillar cell, reticular lamina, and basilar membrane due to outer hair cell motility and due to trans-epithelial fluid pressures. Recent studies of cochlear mechanics show that the active OoC vibrations are much more complicated than classical kinematics models

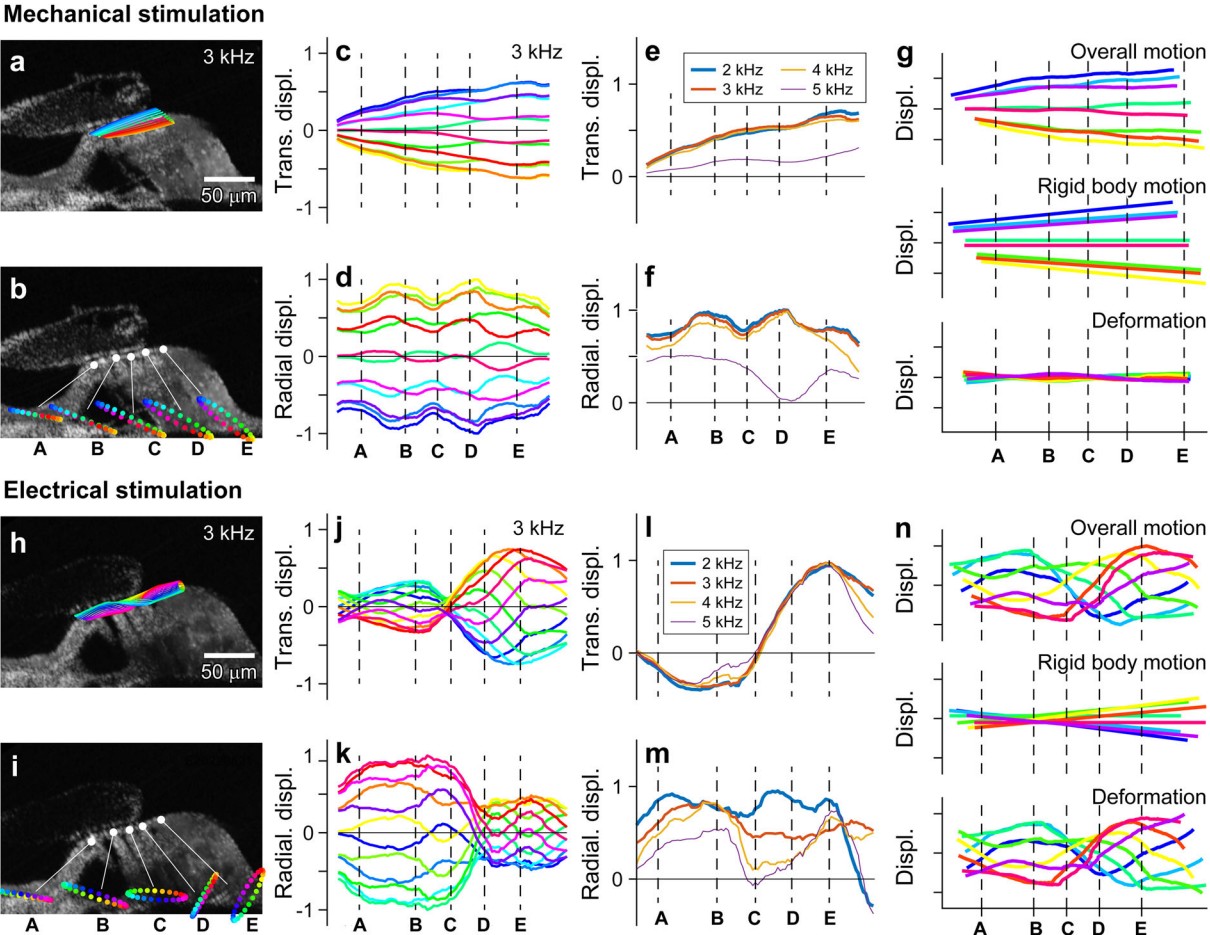

**Fig. 5 | Reticular lamina vibrating patterns. a**, **b** Vibration pattern and trajectories when mechanically stimulated at 3 kHz. The points of interest represent the top of Corti tunnel (A), the top of three outer hair cells (B, C, D), and the lateral end of the reticula lamina (E). **c**, **d** Vibration patterns for the transverse and radial components. **e**, **f** Transverse and radial vibration patterns at four different frequencies. **g** Overall

motion (top, the same as in (**a**) was decomposed into rigid body motion (middle) and deformation (bottom). **h**–**n** Reticular lamina vibrations to electrical stimulations, presented similarly to (**a**–**g**). Data sets used: M0928 and E0621 of year 2022. Animated figure: Supplementary Movie 4 (https://doi.org/10.6084/m9.figshare. 23710614).

suggested. In the following, we first reconcile the classical view of the OoC as a rigid body with the emerging view of deformable OoC. Then, we reassemble our analyses of OoC sub-structures to explain asymmetric force delivery from the outer hair cells.

Previous studies reported that the reticular lamina vibrates like a rigid plate[21,25–27]. Despite lacking in situ observations, the tunnel of Corti formed by the pillar cells has been considered rigid[28–30]. At odds with the rigid body view, recent studies report substantial deformation within the OoC[3,10,11,15,24] which must represent a collective deformation of OoC supporting structures caused by outer hair cell motility. Due to the lack of imaging resolution, however, the origin of OoC compliance could not be specified.

Some theoretical studies predicted the deformable OoC prior to recent empirical observations. For example, Zetes et al. considered both the axial and bending stiffness of the OoC supporting cells based on their ultra-structure, such as the microtubules and the actin crosslinking[8]. Cochlear models based on continuum mechanics have incorporated both the axial and bending stiffness of OoC structures[2,31–33]. It has been suggested that there is an optimal level of OoC frame compliance for the outer hair cells to deliver their power efficiently[4,6,34–36]. Our results support those predictions.

Our results show that the OoC frame is stiff in response to acoustic pressures but compliant with outer hair cell force. This load-dependent behavior is reasonable, considering that a rod can behave as a stiff column or a flexural beam depending on how external forces are applied. Our results show that the reticular lamina and the outer pillar cell behave

either as a rigid bar or as a flexural beam depending on the type of load they bear (Fig. 5 and Fig. 6). The classical kinematics model approximates the OoC motion well when it is subjected to transepithelial pressures (Fig. 2e, f): the reticular lamina (Fig. 5a–g) and the outer pillar cell (Fig. 6a–c) behaves like a stiff truss structure under the M-stim. Meanwhile, when the OoC is subjected to outer hair cell motility, the bending of the reticular lamina (Fig. 5h–n) and the outer pillar cell (Fig. 6d–f) account for the most part of OoC deformation.

A mistaken impression regarding cochlear mechanics is that the basilar membrane is a tough mechanical substrate on which the soft OoC sits. This simplistic view of soft OoC on stiff substrate does not hold toward the apex of the cochlea, where the basilar membrane becomes more compliant than OoC structures, according to existing cochlear models and measurements reviewed in ref. 4. The compliant basilar membrane in the apical cochlea could seem inappropriate as a mechanical reference against which outer hair cells exert force to push and pull the reticular lamina (the stereocilia-side of the OoC).

Our element-by-element analysis of OoC sub-structures was assembled in Fig. 7 to explain how even a compliant basilar membrane can be a proper mechanical reference for active outer hair cells (also see Supplementary Movie 6). Previous studies have described similar ideas[5,30,31]. We considered the OoC deformation when subjected to either trans-epithelial fluid pressure (black arrows, Fig. 7a) or OHC contractile force (red arrow pair, Fig. 7a). When the OoC is subjected to fluid pressure (Fig. 7b), the outer pillar cell acts as a column to push and pull the reticular lamina. The OoC

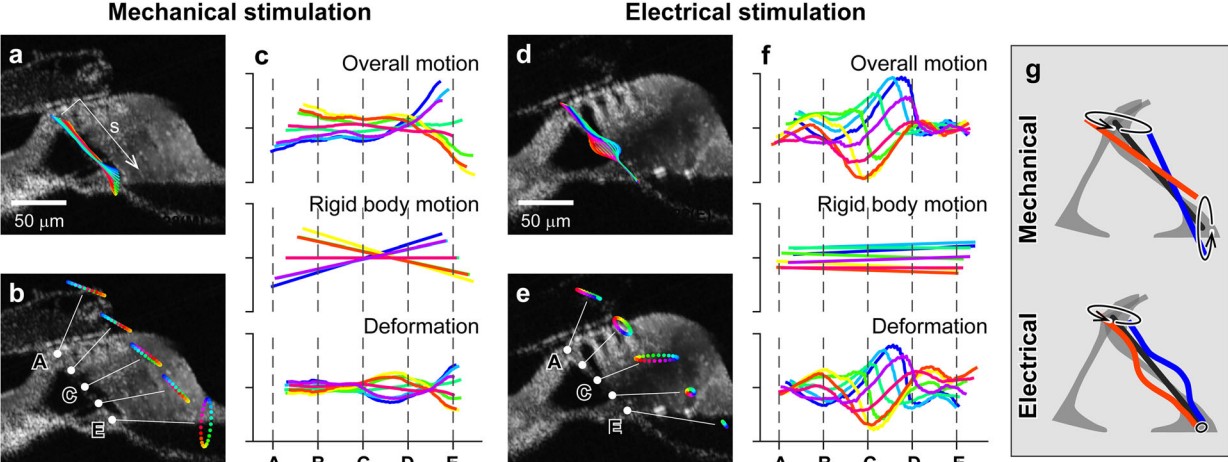

**Fig. 6 | Outer pillar cell vibrating patterns. a**, **b** Vibration pattern and trajectories when mechanically stimulated at 3 kHz. **c** Overall motion (top, the same as in panel **a**) was decomposed into rigid body motion (middle) and deformation (bottom). **d**, **e** Vibration pattern and trajectories when mechanically stimulated at 3 kHz. **f** Overall motion (top, the same as in panel **d**) was decomposed into rigid body motion (middle) and deformation (bottom). **g** Simplified illustrations of outer pillar cell motion due to mechanical and electrical stimulation. The black lines represent the original configuration. The red and blue lines correspond to the largest motions or deformations. Data sets used: M0928 and E0621 of year 2022. Animated figure: Supplementary Movie 5 (https://doi.org/10.6084/m9.figshare.23710599).

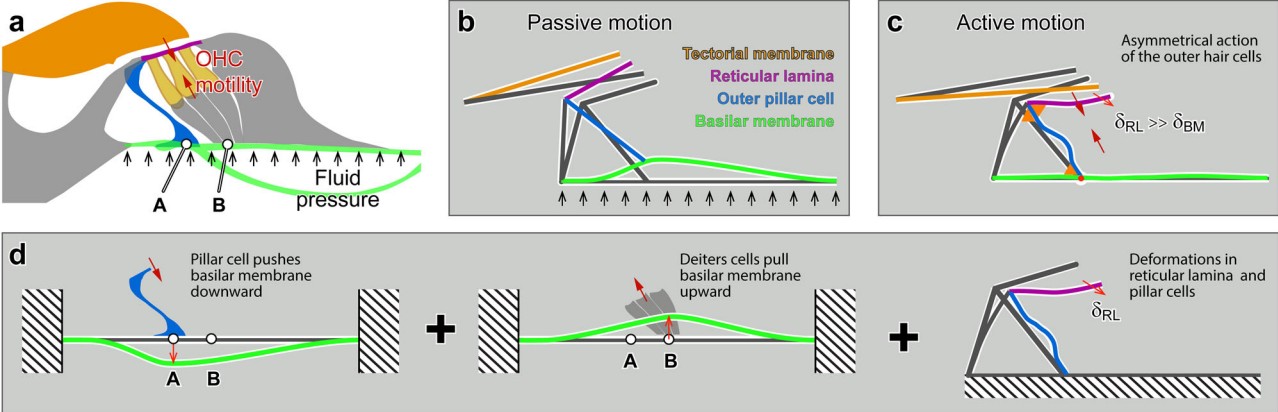

**Fig. 7 | Outer hair cell's leverage. a** The OoC complex vibrates due to external (fluid pressures) and internal (outer hair cell motility) agitations. Four structurally significant structures we investigated in this study were colored differently: yellow tectorial membrane, purple reticular lamina, blue outer pillar cell, and green basilar membrane. Outer hair cell force is delivered to the basilar membrane through two points: 'A' and 'B'. **b** The four structures of interest were represented as deformable sticks for illustrative purposes. The deforming pattern due to fluid pressure was illustrated based on our observations. **c** Deforming pattern due to outer hair cell contraction was drawn based on our observations. **d** A decomposition of the deforming pattern due to outer hair cell contraction. Animated figure: Supplementary Movie 6 (https://doi.org/10.6084/m9.figshare.23710590).

lifts the tectorial membrane with minimal bending of the reticular lamina, which implies (or requires) that the bending stiffness of the tectorial membrane is much smaller than that of the reticular lamina.

Figure 7c illustrates the OoC deformation by outer hair cell motility. Outer hair cells exploit the leverage of the Corti triangle to deflect the top side of the OoC by using the basilar membrane as a mechanical reference. To explain how outer hair cells deflect their top side more than bottom-side ($\delta_{RL} \gg \delta_{BM}$), contracting outer hair cells were represented by a pair of equal and opposite forces (purple arrows in Fig. 7c). In Fig. 7d, force equilibrium in the OoC was decomposed into three diagrams. On the left of panel d, the outer pillar cell is pulled downward by the outer hair cell force applied at the reticular lamina. This force through the outer pillar cell deflects the basilar membrane downward. In the middle of panel d, outer hair cell force is delivered through the Deiters cells to deflect the basilar membrane upward. Due to the proximity between two acting points 'A' and 'B' (for our observed location, 14 percent of the basilar membrane span), the two forces roughly cancel out each other. Instead, the basilar membrane is twisted by the torque made of the force pair, resulting in the

deforming pattern in Fig. 3o. The right of Fig. 7d illustrates the deformation of OoC due to outer hair cell force.

Our observed deformation suggests that the joints of the outer pillar cell with the basilar membrane and the reticular lamina (orange triangles in Fig. 7c) must be stiff, like a bracketed or clamped joint. The proximity between the pillar cell and Deiters cell roots dramatically increases the basilar membrane's effective stiffness felt by the outer hair cells. For example, the bending stiffness increases 340 times when the effective width decreases 7 times, considering that the bending stiffness is proportional to the width to the power of three.

In summary, our results provide a coherent explanation for several aspects of OoC vibrations, such as the greater motion at the reticular lamina than at the basilar membrane in the sensitive cochlea, the higher-order or primary-order vibrations of the basilar membrane, and the rigid-body or bending motion of the reticular lamina. We found that the outer pillar cell acts as the axial column as well as the bending beam for fluid pressures applied to the basilar membrane and for outer hair cell force, respectively. Due to this bending of the pillar cell and the geometrical proximity between

the outer pillar cell and the Deiters cells, the power of the outer hair cells is delivered asymmetrically, greater toward its top side.

An excised cochlea is not sensitive, unlike a healthy live cochlea. For the excised cochlear preparation, we surgically removed parts of the cochlea, including a part of stria vascularis and inter-scala bones. The endolymphatic fluid was replaced with a perilymph-like fluid. Therefore, some characteristics of the live intact cochlea, such as the endocochlear potential and traveling waves, were deprived. We took advantage of these losses as simplifications to investigate the passive and active force transmission within the OoC. For example, hair cell mechano-transduction was abolished due to the loss of endocochlear potential and the calcium level (0.1 mM) in the scala media space. Consequently, the interactions between two vibration types, one caused by fluid pressures and the other induced by outer hair cell motility, were severed in our in vitro preparation. Abolished hair cell mechano-transduction does not mean excised cochlear preparation cannot restore and investigate hair cell mechano-transduction, though. In a similar preparation by Chan and Hudspeth (2005), they had mechano-transduction currents after restoring transepithelial potential and endolymph-like fluid in the scala medial space. For another example, there were no traveling waves due to the wide openings above and beneath the measurement site. While traveling waves are a key characteristic of cochlear physics that shapes frequency tuning, the observed vibrations can complicate interpretation unless measured at a carefully determined orientation (e.g., one scanning line captures different stages of the traveling waves along the top and bottom of the OoC, Frost et al.[18]). Our preparations reduced the longitudinal component of fluid-structure interactions between OoC mechanics and scala fluid dynamics, so we focused on local (2D) OoC mechanics only.

Outer hair cell motility in the present study was induced by the extracellular potential change, unlike both intra and extracellular potential change in live cochlear experiments[37,38]. The outer hair cell electromotility of this study, represented by the reticular laminar motion at the first-row outer hair cell, was about 100 nm/mA at 1 kHz. This value is comparable to other excised gerbil cochlear measurements[27,39]. Although we did not measure transmembrane potential, based on the mechanical properties of the OoC[5], our stimulation of 100 µA at 1 kHz is estimated to be comparable to the outer hair cell transmembrane potential of 3–10 mV in the live cochlea. For example, the OoC stiffness felt by the OHCs near $x = 6.5$ mm is $k_{OoC,OHC} = 75$ mN/m. After considering the typical OoC deformation of 10 nm, and the outer hair motility gain of 0.1 nN/mV, the estimated transmembrane potential is 7.5 mV.

Our in vitro preparation is advantageous for investigating OoC micromechanics--interactions between OoC sub-structures. The mechanistic research questions of our present study do not rely on the compromised physiological conditions. In a nutshell, we report that two sub-structures of OoC (the pillar cell and the reticular lamina) are bent by outer hair cell motility. Our major findings hold despite missing cochlear sensitivity or the traveling waves as long as the OoC structures were preserved. Meanwhile, the challenges of existing in vivo measurements make it difficult to observe what we measured. A notable in vivo study by Cho and Puria partly overcame the resolution challenge[12]. They reported the bending deformation of the reticular lamina in sensitive cochleae. Meanwhile, their observation was one-dimensional and limited to a specific location near the round window (the basal extremity of the cochlea).

## Methods

### Tissue preparation

Mongolian gerbils were used for experiments according to the institutional guidelines of the University Committee on Animal Resources at the University of Rochester (Approval number: 101066/2011-036B). We have complied with all relevant ethical regulations for animal use. Young gerbils of either sex aged 15-30 days old were deeply anesthetized with isoflurane and then decapitated. The cochleae were acutely isolated and placed in a dissection dish filled with artificial perilymph (145 mM sodium gluconate, 7 mM NaCl, 3 mM KCl, 5 mM NaH$_2$PO$_4$, 0.1 mM MgCl$_2$, 5 mM D-glucose, 0.1 mM CaCl$_2$, 5 mM HEPES; pH 7.3-7.4; 300 mOsm). The temperature of

the dissection dish was maintained between 5-10 degrees Celsius. This study targeted a middle-turn location centered at the nominal distance of 6.5 mm from the basal end (Fig. 1a). According to Müller's map[40], this location has the CF of 4 kHz. Hereafter, the CF refers to the CF according to the Müller's map at our measurement location. The nominal distance was determined by translating the number of cochlear turns into distance based on the 3-D coordinates of the basilar membrane of the gerbil cochlea[41]. For example, the location of 6.5 mm corresponds to 1 and 5/8 turns from the apical end (Fig. 1a). In the petri dish, basal and apical turns to the target section were further removed using forceps and sharp blades. At the target location, inter-scala bones were picked away from both basal and apical sides to expose the epithelium (0.8-1 mm-long exposure, Supplementary Fig. 2). Apical and basal openings of the remaining cochlear coil were sealed using cyanoacrylate glue. The reduced cochlear turn was then placed on a plastic disk, with the exposed target region overlying its slit opening (the broken square in Fig. 1b). By applying glue between the bony lateral wall of the preparation and the surface of the plastic disk, the circumference of the excised cochlear turn was sealed and fixed on the disk. The preparation on the plastic disk was then transferred from the dissection dish to the microchamber. It took 60-80 min for tissue preparation.

### Microfluidic chamber

The microfluidic chamber, designed using drawing software and fabricated using stereolithography, was modified from our previous work[20] to enable the flipping preparation described in the following. The chamber had two fluid spaces separated by a plastic disk, where the sample sat (Fig. 1b, illustration not to scale). The top fluid space was open, allowing the coverslip to touch the fluid surface and hence give an undistorted view of the samples. The bottom fluid space was a circular channel that connected four openings: a stimulation port, a release port, and two ports for perfusion. Mechanical stimulations were delivered from a piezoelectric actuator through a silicon rubber placed in the stimulation port. On the other side of the chamber, the release port was functionally analogous to the round window of the intact cochlea. Polytetrafluoroethylene tubing connected to the perfusion ports refreshed fluid in the bottom fluid space. Electrical stimulation across the OoC was applied through a pair of electrodes inserted in the top and bottom fluid spaces, respectively.

### Flipping preparation

The practical depth of field of our OCT imaging was approximately 100 µm. The distance along the optical axis between two extremities of the OoC complex (tectorial membrane attachment point to the spiral limbus and basilar membrane lateral end) was between 150 and 300 µm, depending on the orientation angle. As a result, when the tectorial membrane was in focus, the basilar membrane would be out of focus (Fig. 1d). When the basilar membrane was the target of vibration measurement, the preparation was placed such that the basilar membrane faced upward (called the 'BM-up' preparation, Fig. 1e). Paired magnet arrays embedded in the plastic disk and the chamber top surface ensured consistent placement despite flipping. In a control experiment (Fig. 1f), the OoC vibrations were measured at the TM-up and the BM-up orientations. The angle of the objective was adjusted so that the basilar membrane became horizontal. The relative vibration of the reticular lamina with respect to the basilar membrane (RL re. BM) was measured at different stimulating frequencies. We observed minimal differences between the responses in two orientations (red and black curves in Fig. 1f). That is, the orientations (TM-up or BM-up) did not affect vibrations. The slope angle of the cochlear coil at our measurement location (arctan(dz/ds), where z and s represent the coil elevation and arc axis) did not exceed 10 degrees (mean of 5 degrees in 14 measurements out of 5 cochlear preparations, Supplementary Fig. 2).

### Stimulation and measurements

Electrical stimulation was applied via a pair of Ag/AgCl electrodes. Alternating current was applied by a custom-built current source. The voltage across the tissue was monitored, and the electrical impedance between two

electrodes was typically 5-6 kΩ. Mechanical stimulations were delivered using a piezoelectric actuator (PC4WM, Thorlabs), which was driven by a high-voltage amplifier (E-505.00, Physik Instrumente). Stimulus functions were generated by MATLAB code which also communicates with a data acquisition board (PCI-6353, National Instruments). For vibration measurements, a commercial optical coherence tomography (OCT) imaging system (Ganymede, Thorlabs) was used. The light source of the system had a center wavelength of 900 nm. The A-scan rate was 60 kHz. The system was customized to use a 20X objective (NA 0.4; Mitutoyo) to enhance the resolution in the lateral direction. Full-width at half-maximum resolution was 0.7 and 2.0 μm along the lateral and optical axis, respectively. The OCT system was driven by a custom-written MATLAB program and the vendor-provided function library to perform its M-scan mode for vibration measurements. We ran a set of measurements at 40–60 scanning lines over a radial span of the OoC. The order of the scanning lines was randomized. Stimulations were multi-tone complexes, containing 16-22 frequencies between 1 and 8 kHz (between two octaves below and one above the Müller CF at the nominal location, 4 kHz).

### Two-dimensional analysis

Two-D analysis procedure follows our previous study[5]. We chose the upper collagen band of the basilar membrane as the radial axis. A few dozen M-scans were acquired across an OoC radial section, 2-3 μm apart. Measurements were performed at two incidence angles. For instance, the angles were $\theta_1 = -21$, and $\theta_2 = 9$ in degrees in the example of Fig. 2. The angle θ was adjusted by rotating the optical head of the OCT system with respect to the microchamber. The rotating axis was aligned with the longitudinal axis. Light path obstructed by the lateral wall or the modiolus limited the angular range between 30 and 50 degrees (Δθ in Fig. 1c). The radial and transverse components $d_r$ and $d_t$ of 2-D vibration were calculated from the displacement at two angles.

$$\begin{bmatrix} d_r \\ d_t \end{bmatrix} = \begin{bmatrix} \sin\theta_1 & \cos\theta_1 \\ \sin\theta_2 & \cos\theta_2 \end{bmatrix}^{-1} \begin{bmatrix} d_1 \\ d_2 \end{bmatrix}, \quad (1)$$

where $d_1$ and $d_2$ are the displacement in complex number (having both amplitude and phase information) corresponding to $\theta_1$ and $\theta_2$, respectively.

In Fig. 2a–f, the preparation was placed so that the tectorial membrane faced upward. In such a 'TM-up' preparation, the top side of the OoC was more reflective resulting in better signal-to-noise ratios. Meanwhile, in Fig. 2g–l, the basilar membrane faced upward ('BM-up' preparation) so that we could get clearer optical signals from the basilar membrane. In the example of Fig. 2a, because the focal plane was at the level of the tectorial membrane, the basilar membrane was out-of-focus. Another measurement was made at the focal level near the basilar membrane in this case. The two data sets were combined before 2-D analysis (see Supplementary Fig. 1).

The radial section at the nominal distance of 6.5 mm from the basal end was our plane of interest for 2-D motion analysis. According to previous studies, a clear definition of this 2-D plane has consequences in analyzing the motion components along the anatomical axes of radial, transverse, and longitudinal directions[3,18]. The radial direction of this study is parallel with the flat fiber layer of the basilar membrane's pectinate zone. The transverse direction is normal to the radial and longitudinal directions (Fig. 2e). In practice, the radial section of this study was determined by choosing a scanning line normal to the arc of the cochlear coil. We placed our excised cochlear turns so that the slope along the longitudinal axis at the measurement site was minimal (a mean of 5 degrees in 14 measurements out of 5 cochlea preparations, see a typical longitudinal B-scan image in Supplementary Fig. 2). It was affirmed for every 2-D analysis that measurements at two orientation angles were aligned (Supplementary Fig. 3). Because the scala tube was widely open along the longitudinal direction, the tissue vibrated in-phase over the length so that the motion was predominantly two-dimensional. According to previous studies, the cell's morphology such as the length and thickness corresponds to the mechanical status of the outer

hair cell[42,43]. We monitored the shapes of OoC sub-structures to ensure the structural integrity of our preparations. In our typical preparations, the tissue maintained good morphology at least 3 h from animal death (Supplementary Fig. 4).

### Statistics and reproducibility

This study does not draw any conclusions from statistical analyses. Multiple data sets were presented to demonstrate the reproducibility qualitatively.

### Reporting summary

Further information on research design is available in the Nature Portfolio Reporting Summary linked to this article.

## Data availability

All data and data visualization codes are available through FigShare. https://figshare.com/s/511e2118ad4db66d034c (https://doi.org/10.6084/m9.figshare.25674612). The hyperlinks, DOIs and captions of Supplementary Movies are provided in Supplementary Information.

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

## Acknowledgements

This study was supported by the National Institute of Health of the United States (National Institute on Deafness and Other Communication Disorders, R01 DC014685 and DC020150).

## Author contributions

W.C.L. and J.H.N. designed the research. W.C.L., A.M., J.B., and J.H.N. performed experiments. W.C.L. acquired most data while A.M. and J.H.N. contributed partially. W.C.L. and J.H.N. analyzed the data. J.H.N. created figures, animations, and drafted MS. W.C.L., A.M., and J.H.N. wrote together. Measurement codes were written by J.B., W.C.L., and J.H.N. Analyses codes were written by J.H.N. and W.C.L.

## Competing interests

The authors declare no competing interests.
