## [Peer Review File · Communications Biology]

Reviewers' comments:

Reviewer #1 (Remarks to the Author):

This is a report on a very pressing matter regarding cochlear mechanics. While our view of cochlear mechanics was centred around results obtained from single-point measurements or scanning a proportion of the basilar membrane with laser-interferometers for quite some time, the advent of OCT techniques provided a more comprehensive picture. It was now possible to observe a larger part of the Organ of Corti and its components during acoustic or electrical stimulation.

Results obtained with this technique revolutionised our understanding of cochlear mechanics and it became soon clear that movements of the cochlear partition are far from uniform and that it does not behave as a rigid structure. It is now evident that its elements respond to acoustic stimulation in a quite complex manner, challenging even the idea of the cochlear amplifier in its traditionally assumed function as a cycle-by-cycle amplifier (Cooper, N.P., Vavakou, A. & van der Heijden, M. Vibration hotspots reveal longitudinal funneling of sound-evoked motion in the mammalian cochlea. *Nat Commun* 9, 3054 (2018). <https://doi.org/10.1038/s41467-018-05483-z>).

While it is a fascinating technique, OCT is not without its limitations, and the authors of this paper claim to have solved some of these challenges by employing a post-mortem approach in an excised gerbil cochlea, which allows for a more direct optical access to the Organ of Corti.

As a result of this approach, the authors claim to have been able to dissect passive (caused by direct mechanical stimulation) and active (caused by electrical stimulation of outer hair cells) contributions to organ of Corti movements.

While the results appear sound and are well reported, the question of whether the observations resemble in-vivo movements of the Organ of Corti at all remains.

In a post-mortem preparation such as the one used by the authors, a crucial element of physiological outer hair cells electromotility is missing, namely the physiological, positive endolymphatic potential. It can be assumed that the operating points of outer hair cells are severely shifted away from their resting operating points as the endolymphatic potential approaches 0 mV after excision of the cochlea, making the effects of the electrically stimulated outer hair cells on Organ of Corti movements quite unpredictable (see also below).

I have three main concerns regarding this manuscript:

1. The manuscript lacks an overarching research question. What is the hypothesis here? Is the purpose of this paper to demonstrate that some known features of cochlear mechanics assessed with OCT can also be demonstrated post-mortem?
2. I fail to see the motivation for the post-mortem preparation employed for obtaining the current data set, as it likely causes more problems than it actually solves. The observed results most likely do not resemble physiological responses at all. Results obtained from OCT in alive, sensitive preparation taught

us that the Organ of Corti is much more than just the sum of its elements, and that for instance isolated hair cell responses to voltage commands (Frank, G., W. Hemmert and A. W. Gummer (1999). "Limiting dynamics of high-frequency electromechanical transduction of outer hair cells." *Proc Natl Acad Sci U S A* 96(8): 4420-4425.) might be very different from what they do in-situ (see also more specific comments below). So why go a step back? Most of the advantages the authors claim to see in their post-mortem preparation can also be achieved in a live preparation, with the added benefit that the recordings obtained in such way might closely resemble a physiological situation.

3. I also fail to see the novelty of the reported results- the authors themselves claim several times that their results confirm previous result of others. I do agree that the authors can clearly separate passive (mechanical stimulation) from "active" (electrical stimulation) movements, but this is only because physiological hair cell transduction is disrupted due to fact that this a post-mortem measurement (see also below). What the authors consider an "active response", is the post-mortem response of outer hair cells to electrical stimulation.

Specific comments:

1. The title and the abstract should indicate that this manuscript reports results obtained from a post-mortem preparation.

2. The authors write on p.1, l. 40-41 that their approach "[...]relaxed the difficulties in existing studies". I think it behoves the authors to elaborate on the known difficulties of their own approach, which includes, but is not limited to, the endolymphatic potential which is probably rapidly drifting to values close to 0 mV after excision of the cochlea and this will dramatically alter the active response of outer hair cells to electrical stimulation (see Jacob, S., M. Pienkowski and A. Fridberger (2011). "The Endocochlear Potential Alters Cochlear Micromechanics." *Biophysical Journal* 100(11): 2586-2594.). With ion pumps in outer hair cells (and elsewhere) failing in a post-mortem preparation, it is very likely that the outer hair cells in this preparation are depolarised with consequences for cell turgor pressure, hair bundle position etc. (see Grosh, K., J. Zheng, Y. Zou, E. de Boer and A. L. Nuttall (2004). "High-frequency electromotile responses in the cochlea." *J Acoust Soc Am* 115(5 Pt 1): 2178-2184.). This has not only implications for electrical stimulation of the Organ of Corti, but also for mechanical stimulation.

3. On p. 1, l. 24-26, the authors claim that "[...] the vibrations in the upper lateral part of Organ of Corti show compressive nonlinearity well below the characteristic frequency (CF) [...]", whereas in Strimbu, C. E., L. A. Chiriboga, B. L. Frost and E. S. Olson (2023). "A frame and a hotspot in cochlear mechanics." *bioRxiv*: 2023.2006.2029.547111, it was shown that a large part of the Organ of Corti, including regions within the reticular lamina (RL), at the TM-facing surface of the OHCs does not show compressive non-linearity at sub-BF frequencies. This discrepancy should be recognized in the present manuscript.

4. On p. 1, l. 34-35 the authors claim [...] in live cochlear experiments, it is tricky to isolate active Organ of Corti motion due to outer hair cell motility from passive motion due to acoustic pressures." The authors should probably mention that the application of salicylates greatly reduces prestin-mediated electromotility, rendering the Organ of Corti responses passiv (see Strimbu, C. E. and E. S. Olson (2022).

"Salicylate-induced changes in organ of Corti vibrations." Hearing Research 423: 108389). Thus, it is also possible to dissect passive and active movements in an in-vivo preparation by applying salicylates, a technique which has been used for decades for this purpose.

5. The organisation of the manuscript is also a little unorthodox, as the results section contains substantial parts which belong in my opinion to the discussion section.

Reviewer #2 (Remarks to the Author):

(1) The ms provides good data on possible displacement and vibration modes in the Organ of Corti. It is structured very focussed and the figures are excellent. I also enjoyed very much the movies which give additional information. The ms shows that during electrical stimulation the organ of Corti, in particular the outer pillar cells distort quite a bit and maximal movement is evident not at the level of the basilar membrane but at the reticular lamina. The authors interpret this as consequence of asymmetric force delivery by active mechanical amplification by the OHCs. During mechanical stimulation of the presumably no longer normally functioning Organ of Corti, the responses are different and more linear. The authors interpret this as the 'passive' response of the Organ of Corti. In particular, the very high resolution of their data acquisition and the sophisticated analysis give novel insights into cochlear micromechanics. It would strengthen the work if regarding the validity of their interpretations they address more explicitly also possible weaknesses of their very invasive preparation.

specific comments:

(2) results, page 3, line 11 Could you elaborate more why you assume that you observe passive versus active responses. In the in vivo situation, of course mechanical stimulation would induce active cochlear amplification induced by the receptor potentials. In your preparation, this seems to work no longer, and the OHCs may be non-functional in this respect. Do you think that this is due to the fact that you probably do not have endolymph-like solution on the reticular lamina side of your preparation? Or are there also possible structural changes like a disconnection of the OHC stereocilia from the tectorial membrane possible? Did you, maybe in earlier work, investigate, e.g. with EM techniques, if the stereocilia are still connected or not in your preparation? In the latter case, the vibration patterns could be changed quite a bit. In addition, did you or somebody else try to measure receptor potentials or local microphonics in this type of preparation to assess the functionality or absence of transduction?

(3) Figure 2 The motion arrows in Figure 2K,L (lateral part of OoC) seem to indicate some sort of circular movement. Is this the case? Does it imply some sort of energy focussing or reverberations going on?

(4) page 4, line 1 Your decomposition techniques are quite helpful. Would the larger radial movement components during electrical stimulation (Fig.3) that also seems to be accompanied by local phase changes and severe up and downs of displacement indicate the 'higher modes' you address later on? What about distortions due to excessive power delivery? Here it could be helpful if you try to assess the strength of your electrical stimulation in relation to normally occurring receptor potentials.

(5) page 4, line 16 and below, figure 3,4,5 Your line graphs in Fig. 3G,H,O,P as well as Fig.4.5 are very informative. Did you apply the same normalization for all sub graphs of the same type? That means if I try e.g. to compare mechanical vs. electrical responses in your normalized figures, the differences correspond to similar differences in absolute response magnitude?

(6) Figure 4E,F shows nicely a frequency tuned behaviour of the trans/radial response in the 'passive' situation and an absence of tuning in the 'active' case. Of course motility of single OHCs most likely is not tuned, but since tuning of the whole OoC is determined by the mechanical properties of all structures, in particular the BM, I am surprised that your responses to electrical stimulation do not show at least a little bit of tuning. Could it be that in your case the OHCs are sort of decoupled from the rest of the OoC?

(7) Figure 5,6 provides quite novel data on reticular lamina and pillar cell deformations during electrical stimulation. It really is amazing by how much these supposedly rather rigid structures can bend. If you look at what the OHCs are doing in these situations (e.g. movie 6, 19868_0_video_8000520_s004kt.mp4) this is no longer so surprising. The OHC rows really work hard against each other and not as a unity, thereby exerting stress to the reticular lamina and the outer pillar. If I were to build a mechanical amplifier, I would take care that no energy is lost in such internal most likely cancellation movements. How do you interpret this?

(8) discussion is straight forward and focussed. I just would have appreciated more in depth assessment of the possible pitfalls and artefacts of such a preparation. I have some doubts if the behaviour of the electrically stimulated OHCs would correspond to what happens normally in vivo. But this does not diminish the value of your excellent data in terms of possible interactions in the OoC.

(9) For me another interesting discussion topic (also relevant for abstract,title) would be: The 'force transmission' could be less 'asymmetric' than you imply. The force produced by OHC movement by contractions and elongations along their length axis initially should be of similar magnitude in both directions and obviously translates into larger deflections at the top of the OHCs. At the bottom, towards the supporting cells and basilar membrane the force may increase the stiffness of those structures. Stiffness (which has the dimension of a force) may be an important component of cochlear amplification and also of increase of cochlear tuning sharpness.

(10) tissue preparation Could you give the longitudinal extend of your excised OoC prep? How many longitudinal OHCs are in your preparation? Longitudinal interactions are also quite important for tuning and possibly for the distribution of motion energy coming from the OHCs.

(11) Stimulation: What were the voltage and the voltage steps you applied? Please try to give some assessment what this could imply for transmembrane voltage across OHCs.

Reviewer #3 (Remarks to the Author):

The authors present a paper based on well conducted experiment on an in vitro preparation of the 4kHz frequency region of the gerbil cochlea using cutting edge techniques to address and attempt to resolve conflicting outcomes from recent papers using similar techniques on largely in vivo preparations. In general, the paper is well written, and the data are presented clearly. However, there is almost no distinction made throughout the paper between in vivo and in vitro measurements, between measurements made at different frequency regions, and between measurements and theory. These distinctions are important because, in addition to its complex cellular architecture, which differs in a graded manner along the cochlea, the cochlea has a complex electrochemistry that strongly determines its electromechanical properties and performance. These properties are often neglected in theoretical papers and can be compromised in vitro studies. The majority of comments below are directed at these considerations.

Page 2 Abstract

Line 5. "recent data". Please specify the type of data, the reviewer believes it to be in vivo data.

Line 7. Please replace "we present motion measurements" with "We present motion measurements from an isolated in vitro preparation"

Page 3 Introduction

Line 7. Please replace "the hair cells" with "the hair cells as indicated by recent theoretical models"

Lines 8-9 "Meanwhile, the OoC is flexible enough to get deformed by motile outer hair cells." Is this based on measurements or models?

Line 9. "Meanwhile, the OoC is flexible enough to get deformed by motile outer hair cells." Please support these two important statements above by references to direct measurements. If these are not available, then by references to theoretical studies.

Lines 34- 35 "First, in live cochlear experiments, it is tricky to isolate active OoC motion due to outer hair cell motility from passive motion due to acoustic pressures." Please state the basis for this statement. For example, it is possible to target prestin without changing the passive mechanical properties of OHCs and to remove the influence of cochlear amplification and compression, thereby causing the altered mechanical and electrical responses of the cochlea to be linear over their entire Deiters' level and frequency range (e.g. Dallos et al., <https://doi.org/10.1016/j.neuron.2008.02.028> , Weddell et al 10.1016/j.cub.2011.08.001). It has however, been difficult both in vivo and in vitro to isolate Deiters' cell activity from OHC motility without selectively targeting Deiters' cells e.g. Zhao HB et al Neurophysiol. 2022 Jan 1;127(1):313-327. doi: 10.1152/jn.00468.2021. Lukashkina doi: 10.1523/JNEUROSCI.2127-21.2022

Line 40. "relaxed". Perhaps try "avoided", or perhaps "addressed"?

Lines 41-42. "By reducing the bones in the optical path, the vibrometry image "

Perhaps say something like: "By removing structures that would normally obstruct the optical path...."

Introduction page 2

Lines 3-4 "We focused on 2-D motions of the basilar membrane, reticular lamina, and outer pillar cell

because they construct the frame for active and passive force transmission in the OoC.”

This is fine for studying radial and transfer motion, but not for longitudinal motion. Perhaps specify radial and transverse, because others have shown that longitudinal flow of energy plays important roles in shaping the active and passive mechanical properties of the OoC.

Lines 5-7. “Our results provide a coherent explanation of how outer hair cells exploit their structural frame to enhance vibrations of the upper part of the OoC while minimizing interference with the basilar membrane.”

This could be misleading and controversial as it is written. It might be more accurate to say “Our results provide a coherent explanation of how outer hair cells exploit their structural frame to enhance vibrations of the upper part of the OoC relative to those of the basilar membrane.”

Results

Page 3 lines 6-8. Why didn't you include Deiters' cells in your study? There are *in vivo* and *in vivo* studies that show their activity influences OHC motility, they can control cochlear sensitivity, and have been identified as the essential elements of feedforward models of longitudinal force transfer in the OoC. e.g. Yoon et al., *Biophysical Journal* 100:1-10.

Lines 11-12. “of passive (dead or insensitive) and active (sensitive) cochlear responses of the intact cochlea, respectively (Fig. 2).” What are the criteria for categorizing “active” and “dead”. There appears to be nothing about this in the methods. It is important that the reader understands the criteria for this distinction.

Lines 13-14. “In contrast, the vibrating pattern was non-monotonic for E stim (Fig. 2H and J).”

This conclusion involves some unknowns over which you have no control. These include: Mechanical stimulation may or may not cause OHC electromotility depending on the state of the preparation and the stimulus frequency relative to the frequency of the measurement location. Electrical stimulation could drive both OHC electromotility and Deiters cell electromotility, as demonstrated directly *in vitro* (Yu, Zhau *PLoS ONE* 4: e7923) depending on the state of the preparation. This latter possibility could cause changes to the OHC structural cage that are not encountered *in vivo*. Can you please comment on this because it is important for understanding and interpreting your findings. One way out of this apparent dilemma is to suggest that during electrical stimulation, Deiters' cell motility may always accompany OHC motility (e.g. Lukashkina et al, *Journal of Neuroscience* 42:5660-71).and modify OHC motility (e.g. Yu, Zhau *PLoS ONE* 4: e7923). With mechanical stimulation, the mechanics and the electrochemistry of the preparation has been altered so that it resembles a passive cochlea.

Lines 18-19. “The motion trajectories were transversely dominant in M-stim responses, while E-stim responses showed greater radial motions, especially around outer hair cells.”

Radial vibration of the BM may vary with species and frequency place on the BM and it is thought this point should be made clear. It would be interesting to compare these parameters in the 70 kHz region of the mouse cochlea with the 4 kHz region in the gerbil.

Page 4. Lines 1-14. This section deals with several major issues that the reviewer believes should be

addressed to make your findings more useful in understanding the electromechanical properties of the cochlear partition.

The first of these is that the structure of the cochlear partition. There are many references available describing the structure of the cellular components and of the structural frame encasing the sensory electromotile outer hair cells, and indeed the OHCs themselves, that show strong structural differences from apex to base. These differences are reflected in the *in vivo* physiology and has led to the suggestion that apical regions of the cochlea process isotonic forces, while the basal region processes isometric force (<https://doi.org/10.1038/s41598-017-04279-3>). Thus, the transvers and radial displacements that you very elegantly demonstrate in your paper, may not be replicated in more basal regions and it is recommended that you stated this, or something to that effect, because it avoids criticism and paves the way for further measurements to address this question.

The second concerns the interpretation of the outcomes from Mechanical and Electrical stimulation of the preparation (see Methods). The outcomes from both forms of stimulation depend on the physiological state of the preparations, which might not be ideal considering the mechanical trauma, the time it takes to make the preparation before measurement, and the substitution of endolymph for perilymph. The sensitivity of *in vivo* preparations depends critically on the sensitivity of the cochlea. A difference in sensitivity of 5dB between preparations can alter the mechanical and electrical properties of the cochlea, for example in terms of saturation and nonlinearity and in the mechanical responses of the cochlear to acoustic stimulation, which may also account for differences in the reported mode of vibration across the width of the cochlea (P4,L38-39). Thus, in response to mechanical stimulation, the preparation may be that of a passive cochlea, without amplification. Electrical stimulation may cause the OHCs to change length, as appears in Figs. 4-6. In addition, it Deiters' cells, at least, are motile (e.g. Dulon et al, 1994 doi: 10.1006/bbrc.1994.1841) and could contribute to the complex motion of the cochlear partition during electrical stimulation, whereas they may contribute indirectly by influencing OHC motility during low-level mechanical stimulation *in vivo* (Lukashkina et al., 2022). Thus, the responses of the structural framework surrounding the OHCs to electrical stimulation may resemble those measured *in vivo* in sensitive cochlea to mechanical stimulation, but perhaps with contributions from elements, such as the Deiters' cells, which would not normally contribute to the movement of the structural framework during near threshold mechanical stimulation. It is recommended that the authors revise this section and take into account the points raised above.

Dear Reviewers:

The authors appreciate the thoughtful critiques of the reviewers. In response to the reviewers' shared comment, we added a new section in the Discussion titled "Limitations of in vitro cochlear mechanics measurements". Because the new section applies to multiple comments, we present the new section in the following (in red color) before responding point-by-point (in blue color).

An excised cochlea is not sensitive, unlike a healthy live cochlea. For the excised cochlear preparation, we surgically removed parts of the cochlea, including the stria vascularis and inter-scala bones. The endolymphatic fluid was replaced with a perilymph-like fluid. Therefore, some characteristics of the live intact cochlea, such as the endocochlear potential and traveling waves, were deprived. We took advantage of these losses as simplifications to investigate the passive and active force transmission within the OoC. For example, hair cell mechano-transduction was abolished due to the loss of endocochlear potential and the calcium level (0.1 mM) in the scala media space. Consequently, the interactions between two vibration types, one caused by fluid pressures and the other induced by outer hair cell motility, were severed in our in vitro preparation. Abolished hair cell mechano-transduction does not mean excised cochlear preparation cannot restore and investigate hair cell mechano-transduction, though. In a similar preparation by Chan and Hudspeth (2005), they had mechano-transduction currents after restoring transepithelial potential and endolymph-like fluid in the scala medial space. For another example, there were no traveling waves due to the wide openings above and beneath the measurement site. While traveling waves are a key characteristic of cochlear physics that shapes frequency tuning, the observed vibrations can complicate interpretation unless measured at a carefully determined orientation (e.g., one scanning line captures different stages of the traveling waves along the top and bottom of the OoC, Frost, Strimbu, 2023). Our preparations reduced the longitudinal component of fluid-structure interactions between OoC mechanics and scala fluid dynamics, so we focused on local (2D) OoC mechanics only.

Outer hair cell motility in the present study was induced by the extracellular potential change, unlike both intra and extracellular potential change in live cochlear experiments (Levic, Lukashkina et al. 2022, Lukashkina, Levic et al. 2023). The outer hair cell electromotility of this study, represented by the reticular laminar motion at the first-row outer hair cell, was about 100 nm/mA at 1 kHz. This value is comparable to other excised gerbil cochlear measurements (Chan and Hudspeth 2005, Karavitaki and Mountain 2007). Although we did not measure transmembrane potential, based on mechanical properties of the OoC (Zhou, Jabeen et al. 2022), our stimulation of 100 μ A at 1 kHz is estimated to be comparable to the outer hair cell transmembrane potential of 3-10 mV in the live cochlea. For example, the OoC stiffness felt by the OHCs near $x = 6.5$ mm is $k_{\text{OoC,OHC}} = 75$ mN/m. After considering the typical OoC deformation of 10 nm, and the outer hair motility gain of 0.1 nN/mV, the estimated transmembrane potential is 7.5 mV.

Our in vitro preparation is advantageous for investigating OoC micromechanics--interactions between OoC sub-structures. The mechanistic research questions of our present study do not rely on the compromised physiological conditions. In a nutshell, we report that two sub-structures of OoC (the pillar cell and the reticular lamina) are bent by outer hair cell motility. Our major findings hold despite missing cochlear sensitivity or the traveling waves as long as the OoC structures were preserved. Meanwhile, the challenges of existing in vivo measurements make it difficult to observe what we measured. A notable in vivo study by Puria and his colleagues partly overcame the resolution challenge. They reported the

bending deformation of the reticular lamina in sensitive cochleae. Meanwhile, their observation was one-dimensional and limited to a specific location near the round window (the basal extremity of the cochlea).

Reviewers' comments:

Reviewer #1 (Remarks to the Author):

This is a report on a very pressing matter regarding cochlear mechanics. While our view of cochlear mechanics was centred around results obtained from single-point measurements or scanning a proportion of the basilar membrane with laser-interferometers for quite some time, the advent of OCT techniques provided a more comprehensive picture. It was now possible to observe a larger part of the Organ of Corti and its components during acoustic or electrical stimulation.

Results obtained with this technique revolutionised our understanding of cochlear mechanics and it became soon clear that movements of the cochlear partition are far from uniform and that it does not behave as a rigid structure. It is now evident that its elements respond to acoustic stimulation in a quite complex manner, challenging even the idea of the cochlear amplifier in its traditionally assumed function as a cycle-by-cycle amplifier (Cooper, N.P., Vavakou, A. & van der Heijden, M. Vibration hotspots reveal longitudinal funneling of sound-evoked motion in the mammalian cochlea. Nat Commun 9, 3054 (2018). <https://doi.org/10.1038/s41467-018-05483-z>).

While it is a fascinating technique, OCT is not without its limitations, and the authors of this paper claim to have solved some of these challenges by employing a post-mortem approach in an excised gerbil cochlea, which allows for a more direct optical access to the Organ of Corti.

As a result of this approach, the authors claim to have been able to dissect passive (caused by direct mechanical stimulation) and active (caused by electrical stimulation of outer hair cells) contributions to organ of Corti movements.

While the results appear sound and are well reported, the question of whether the observations resemble in-vivo movements of the Organ of Corti at all remains.

In a post-mortem preparation such as the one used by the authors, a crucial element of physiological outer hair cells electromotility is missing, namely the physiological, positive endolymphatic potential. It can be assumed that the operating points of outer hair cells are severely shifted away from their resting operating points as the endolymphatic potential approaches 0 mV after excision of the cochlea, making the effects of the electrically stimulated outer hair cells on Organ of Corti movements quite unpredictable (see also below).

Thank you for acknowledging the sound presentation of our results. As we acknowledged in the MS, some physiological characteristics, such as the endolymphatic potential, disappear as soon as the animal is killed. On the other hand, structural mechanics, such as the force and deformation relationship, were minimally compromised within the time scale of our experiment. For instance, the vibration amplitudes

and patterns are highly consistent over a few hours (Jabeen et al., 2020). We made an additional effort not to confuse our questions or arguments with those of live animal studies.

Mechanics (vibrations) due to outer hair cell motility and fluid pressures are the subject of this paper. The subjects directly relevant to cochlear sensitivity or mechano-transduction (MET) are not the scope of this work. In many ongoing debates regarding hearing sensitivity, mechanics (active and passive force transmission in the OoC) play a central role in the debates. Some OoC micromechanics cannot be readily explored in live cochleae because the motions of OoC fine structures are difficult to capture through temporal bones and because the view angle and accessible locations are inevitably limited.

We agree with the reviewer that the hair cells' resting open probability, thus the resting MET current, must be substantially shifted in our preparation from the physiological state. The MET current must be minimal in our preparation. This abolishment of hair cell MET helps to distinguish the passive and active vibrations more clearly. If our preparations had MET current, it would not be possible to isolate the motions due to active feedback from the motions due to fluid pressures, similar to existing live animal experiments.

I have three main concerns regarding this manuscript:

1. The manuscript lacks an overarching research question. What is the hypothesis here? Is the purpose of this paper to demonstrate that some known features of cochlear mechanics assessed with OCT can also be demonstrated post-mortem?

Our study was objective-driven rather than hypothesis-driven. That said, our objective (to characterize active and passive force transmission in the OoC, Lines 39-40 in the original MS) can readily be translated into a few hypotheses. Our hypothesis can be that active force transmission in the OoC induces bending deformation in the OPC and the RL, unlike passive force transmission. This hypothesis has been alluded to in recent live animal experiments but could not be clearly concluded.

2. I fail to see the motivation for the post-mortem preparation employed for obtaining the current data set, as it likely causes more problems than it actually solves. The observed results most likely do not resemble physiological responses at all. Results obtained from OCT in alive, sensitive preparation taught us that the Organ of Corti is much more than just the sum of its elements, and that for instance isolated hair cell responses to voltage commands (Frank, G., W. Hemmert and A. W. Gummer (1999). "Limiting dynamics of high-frequency electromechanical transduction of outer hair cells." Proc Natl Acad Sci U S A 96(8): 4420-4425.) might be very different from what they do in-situ (see also more specific comments below). So why go a step back? Most of the advantages the authors claim to see in their post-mortem preparation can also be achieved in a live preparation, with the added benefit that the recordings obtained in such way might closely resemble a physiological situation.

We placed our response upfront, but responding further to some specific comments would be good. As the reviewer said, sensitive cochlear responses are nonlinear. The feedback loop responsible for the nonlinearity creates intrinsic difficulty in separating the responses due to outer hair cell motility. For instance, subtracting salicylate-treated OoC vibrations from intact responses does not result in the OoC vibrations due to outer hair cell motility.

The reviewer brought up a great example. Frank et al.'s paper does not show the responses of sensitive cochlea, not even in situ responses of outer hair cells. As the reviewer said, the OHC action observed by Frank et al. must be different from outer-hair-cell operation in situ. However, because the underlying physical principle holds (in their case, the relationship between transmembrane potential and cell motility), their data still inspire researchers. The outer hair cells in our preparation are under compromised (arguably controlled) physiological conditions, but physics principles (force-displacement relationship) should hold whether the cochlea is sensitive or not. In live animal experiments, few ways are known to extract the vibrations of OoC sub-components due to outer hair cell motility. Therefore, our data must be useful in illuminating OoC deformation patterns due to the in situ action of outer hair cell motility.

3. I also fail to see the novelty of the reported results- the authors themselves claim several times that their results confirm previous result of others. I do agree that the authors can clearly separate passive (mechanical stimulation) from "active" (electrical stimulation) movements, but this is only because physiological hair cell transduction is disrupted due to fact that this a post-mortem measurement (see also below). What the authors consider an "active response", is the post-mortem response of outer hair cells to electrical stimulation.

Yes, we could isolate the motion due to outer hair cell motility because the cochlear sensitivity was abolished. We understand the reviewer wants us to more clearly state that the hair cell MET was disrupted in our preparation. We made the point clearer in the Discussion.

Specific comments:

1. The title and the abstract should indicate that this manuscript reports results obtained from a post-mortem preparation.

The title has been changed for clarification.

*2. The authors write on p.1, l. 40-41 that their approach "[..]relaxed the difficulties in existing studies". I think it behoves the authors to elaborate on the known difficulties of their own approach, which includes, but is not limited to, the endolymphatic potential which is probably rapidly drifting to values close to 0 mV after excision of the cochlea and this will dramatically alter the active response of outer hair cells to electrical stimulation (see Jacob, S., M. Pienkowski and A. Fridberger (2011). "The Endocochlear Potential Alters Cochlear Micromechanics." *Biophysical Journal* 100(11): 2586-2594.). With ion pumps in outer hair cells (and elsewhere) failing in a post-mortem preparation, it is very likely that the outer hair cells in this preparation are depolarised with consequences for cell turgor pressure, hair bundle position etc. (see Grosh, K., J. Zheng, Y. Zou, E. de Boer and A. L. Nuttall (2004). "High-frequency electromotile responses in the cochlea." *J Acoust Soc Am* 115(5 Pt 1): 2178-2184.). This has not only implications for electrical stimulation of the Organ of Corti, but also for mechanical stimulation.*

Yes, our preparation took advantage of the abolish the hair cell MET, partly due to abolished endocochlear potential. This point was made clear by adding a section in the Discussion.

3. On p. 1, l. 24-26, the authors claim that "[...] the vibrations in the upper lateral part of Organ of Corti

show compressive nonlinearity well below the characteristic frequency (CF) [...], whereas in Strimbu, C. E., L. A. Chiriboga, B. L. Frost and E. S. Olson (2023). "A frame and a hotspot in cochlear mechanics." *bioRxiv*: 2023.2006.2029.547111, it was shown that a large part of the Organ of Corti, including regions within the reticular lamina (RL), at the TM-facing surface of the OHCs does not show compressive nonlinearity at sub-BF frequencies. This discrepancy should be recognized in the present manuscript. Strimbu et al.'s results might agree with our observation at the inner-most part of the RL, but not with that at the outer-most RL. An important point of our observation (and Cho & Puria's) is that the RL cannot be represented by a single pixel (or DoF). The Strimbu et al. work is a good example that our work can provide context for, we will cite their work as their work is published.

4. On p. 1, l. 34-35 the authors claim [...] in live cochlear experiments, it is tricky to isolate active Organ of Corti motion due to outer hair cell motility from passive motion due to acoustic pressures." The authors should probably mention that the application of salicylates greatly reduces prestin-mediated electromotility, rendering the Organ of Corti responses passive (see Strimbu, C. E. and E. S. Olson (2022). "Salicylate-induced changes in organ of Corti vibrations." *Hearing Research* 423: 108389). Thus, it is also possible to dissect passive and active movements in an in-vivo preparation by applying salicylates, a technique which has been used for decades for this purpose.

We disagree with this salicylate argument for three reasons. First, the active OoC motion due to outer hair cell motility cannot be isolated from the motion of the sensitive cochlea. Passive mechanics can be observed from ex vivo cochlea without using drugs such as salicylate or furosemide. As the reviewer mentioned above, active motion due to outer hair cell motility cannot be obtained from the difference between the motions of sensitive and insensitive cochlea. Second, it is difficult to suppress outer hair cell motility completely by salicylic acid. When an excess amount of drug for complete abolishment is applied, it causes irreversible damage to the outer hair cells and animals become unstable by an overdose. In systemic delivery, the extent and the timing of suppression and recovery due to salicylic acid vary over the cochlear length. This issue is not less significant in local (round window) delivery. Near the delivery site, irreversible damage is caused by the salicylate. Far from the delivery site, outer hair cells are unaffected by the drug. Applying salicylic acid is a convenient means to reduce outer hair cells' action, but it is not a clean control (on or off) of their motility. Finally, the purposes of live animal experiments and the present experiment are different. The change due to salicylate application at the cochlear systems level is used to investigate the change in cochlear sensitivity/amplification due to suppression of outer hair cell motility. The focus of the present study is at the level of OoC micromechanics, not at the level of cochlear mechanics. Our approach does not and cannot discuss cochlear sensitivity. We investigate force transmission from the outer hair cells to their framing structures.

5. The organisation of the manuscript is also a little unorthodox, as the results section contains substantial parts which belong in my opinion to the discussion section.

We added a paragraph to the Discussion.

Reviewer #2 (Remarks to the Author):

(1) The ms provides good data on possible displacement and vibration modes in the Organ of Corti. It is

structured very focussed and the figures are excellent. I also enjoyed very much the movies which give additional information. The ms shows that during electrical stimulation the organ of Corti, in particular the outer pillar cells distort quite a bit and maximal movement is evident not at the level of the basilar membrane but at the reticular lamina. The authors interpret this as consequence of asymmetric force delivery by active mechanical amplification by the OHCs. During mechanical stimulation of the presumably no longer normally functioning Organ of Corti, the responses are different and more linear. The authors interpret this as the 'passive' response of the Organ of Corti. In particular, the very high resolution of their data acquisition and the sophisticated analysis give novel insights into cochlear micromechanics. It would strengthen the work if regarding the validity of their interpretations they address more explicitly also possible weaknesses of their very invasive preparation.

We thank the reviewer for appreciating our results. We added a paragraph in the Discussion regarding the differences between intact and excised cochlear measurements.

specific comments:

(2) results, page 3, line 11 Could you elaborate more why you assume that you observe passive versus active responses. In the in vivo situation, of course mechanical stimulation would induce active cochlear amplification induced by the receptor potentials. In your preparation, this seems to work no longer, and the OHCs may be non-functional in this respect.

We added a new paragraph in the Discussion to address this comment. We believe we understand the reviewer's point. Whether we call it functional or non-functional may depend on which function we are interested in. If we are interested in cochlear MET, our prepared OoC is considered non-functional. If we are interested in structural mechanics or in outer hair cell motility, our prepared OoC is considered functional. Severing the outer hair cells' active feedback loop by abolishing their MET enabled us to observe the mechanics due to only outer hair cell motility.

Do you think that this is due to the fact that you probably do not have endolymph-like solution on the reticular lamina side of your preparation?

In the present study, we used the perilymph-like solution for both the scala media and the scala tympani spaces because of the flip protocol. In our previous study (Jabeen et al. 2020), we applied an endolymph-like solution in the scala media space. Mechanics were hardly affected by the endolymph-perilymph condition. The hair cell MET was minimized for three reasons. First, there was no endolymphatic potential. Second, the calcium level in the scala medial space is higher than the physiological level, which reduces the open probability of hair cell MET. Finally, in the excised cochlea, the membrane potential of hair cells is likely lost. Unfortunately, we do not have firm evidence in either way for now. Albeit preliminary, we've observed OoC microphonic potentials when the transepithelial potential (like the endocochlear potential) and the endolymph-like solution were restored. The microphonic potentials did not last long, however (< 1.5 hours from decapitation). Although we did not measure the membrane potential of outer hair cells, considering the time scale of our measurements (1-3 hours from decapitation), a large fraction of membrane potential could have been lost.

Or are there also possible structural changes like a disconnection of the OHC stereocilia from the tectorial membrane possible? Did you, maybe in earlier work, investigate, e.g. with EM techniques, if the

stereocilia are still connected or not in your preparation? In the latter case, the vibration patterns could be changed quite a bit.

The tectorial membrane remained attached in the results presented in this study. In our previous paper (Jabeen et al., 2020), we reported dedicated data to demonstrate the integrity of the tectorial membrane attachment to the OoC. Indeed, this attachment is highly fragile. Fortunately, our imaging provides fine enough resolution to detect the sub-TM gap size change confidently, thus its detachment from the OoC. We used the data with the tectorial membrane confidently attached to the OoC. We showed that the OoC motion, especially the relative motion between the tectorial membrane and the reticular lamina, changes depending on the tectorial membrane integrity and its attachment to the reticular lamina. This study reports only intact tectorial membrane cases.

In addition, did you or somebody else try to measure receptor potentials or local microphonics in this type of preparation to assess the functionality or absence of transduction?

With a similar excised cochlear preparation as ours, Chan and Hudspeth (2005) reported MET current when they applied a trans-epithelial electric potential comparable to the endocochlear potential. Although not the scope of this present study, we also observed microphonic potential when we applied (restored) the trans-epithelial potential and the endolymph. However, for the purpose of this study (isolating active and passive motions) because MET current confounds the results due to feedback action, we did not pursue restoring hair cell MET.

(3) Figure 2 The motion arrows in Figure 2K,L (lateral part of OoC) seem to indicate some sort of circular movement. Is this the case? Does it imply some sort of energy focussing or reverberations going on?
Yes, we observed the swirling pattern in 'Active' vibrations. We believe that it suggests an action of area motor (Shokrian et al., 2020; Guinan 2022). This observation, though interesting, is not aligned with the theme of the present paper, and we are not ready for discussing its consequences.

(4) page 4, line 1 Your decomposition techniques are quite helpful. Would the larger radial movement components during electrical stimulation (Fig.3) that also seems to be accompanied by local phase changes and severe up and downs of displacement indicate the 'higher modes' you address later on? What about distortions due to excessive power delivery? Here it could be helpful if you try to assess the strength of your electrical stimulation in relation to normally occurring receptor potentials.

In this work, we abstain from discussing the higher modes because a higher-order mode depends on the type (not amplitude, either M-stim or E-stim) of applied force. The simple vibrating pattern of Fig. 3G is the primary mode when subjected to fluid pressures. Meanwhile, the complex vibrating pattern of Fig. 3O must be the primary mode when the OoC is vibrated by outer hair cell motility, considering that the E-stim vibrating pattern remains similar despite frequencies.

Distortions can occur when hair cell MET saturates. As the reviewer wrote, unlike the physiological condition where outer hair cell motility is induced by MET currents, in our preparations, outer hair cell motility was caused by extracellular potential changes. Under our condition, we could not increase the current level to the saturation of outer hair cell motility because an excess current (> 1 mA) tended to cause cell damage. Regarding estimating equivalent receptor potential, we added our speculation in the Discussion.

(5) page 4, line 16 and below, figure 3,4,5 Your line graphs in Fig. 3G,H,O,P as well as Fig.4.5 are very informative. Did you apply the same normalization for all sub graphs of the same type? That means if I try e.g. to compare mechanical vs. electrical responses in your normalized figures, the differences correspond to similar differences in absolute response magnitude?

Yes. The vibration components were normalized by the peak component (either transverse or radial) motion amplitude of the respective simulation types. Therefore, as the reviewer correctly understood, the ratio (or difference) between transverse and radial components can be read from those plots. We added clarifying statements in a Results section.

(6) Figure 4E,F shows nicely a frequency tuned behaviour of the trans/radial response in the 'passive' situation and an absence of tuning in the 'active' case. Of course motility of single OHCs most likely is not tuned, but since tuning of the whole OoC is determined by the mechanical properties of all structures, in particular the BM, I am surprised that your responses to electrical stimulation do not show at least a little bit of tuning. Could it be that in your case the OHCs are sort of decoupled from the rest of the OoC?

We thank the reviewer for bringing this up. The peak frequency shown in Fig. 4E does not correspond to the best-responding frequency. Had we drawn a mechanical gain (m/Pa) versus frequency to observe a peak, that could be called a BF. The frequency-dependent curve of the M-stim response in Fig. 4E represents a change in the vibrating pattern (eigenvector or mode). The results in Fig. 4D should be interpreted as the change in vibrating patterns over frequency for the M-stim. Unlike the case of M-stim, the pattern remained similar despite varying frequencies in the case of E-stim.

This point was clarified in the revised MS.

(7) Figure 5,6 provides quite novel data on reticular lamina and pillar cell deformations during electrical stimulation. It really is amazing by how much these supposedly rather rigid structures can bend. If you look at what the OHCs are doing in these situations (e.g. movie 6, 19868_0_video_8000520_s004kt.mp4) this is no longer so surprising. The OHC rows really work hard against each other and not as a unity, thereby exerting stress to the reticular lamina and the outer pillar. If I were to build a mechanical amplifier, I would take care that no energy is lost in such internal most likely cancellation movements. How do you interpret this?

It is great to hear that our movie helped. Yes, what the reviewer said is the highlight of our finding—the outer hair cells bend the framing structures of the OoC.

We appreciate this insightful question from the reviewer. Although the energy loss dilemma raised by the reviewer is beyond the scope of this paper, here we share our speculations and ongoing efforts. We are currently trying to reconcile our computer model with our empirical observations. Bending (compliant) structures might not be ideal force transmission media. Notably, the non-structural space in OoC is filled with an incompressible fluid. If the Corti fluid acts as a medium for 'active' force transmission, the outer hair cells might take advantage of the OoC as a peristaltic tube for power transmission (Zagadou, Mountain, 2012; van der Heijden, 2014; Shokrian et al., 2020).

(8) discussion is straight forward and focussed. I just would have appreciated more in depth assessment

of the possible pitfalls and artefacts of such a preparation. I have some doubts if the behaviour of the electrically stimulated OHCs would correspond to what happens normally in vivo. But this does not diminish the value of your excellent data in terms of possible interactions in the OoC.

We added a new paragraph in the Discussion.

(9) For me another interesting discussion topic (also relevant for abstract,title) would be: The 'force transmission' could be less 'asymmetric' than you imply. The force produced by OHC movement by contractions and elongations along their length axis initially should be of similar magnitude in both directions and obviously translates into larger deflections at the top of the OHCs. At the bottom, towards the supporting cells and basilar membrane the force may increase the stiffness of those structures. Stiffness (which has the dimension of a force) may be an important component of cochlear amplification and also of increase of cochlear tuning sharpness.

This is a fair point. We admit that the wording asymmetry is used loosely in terms of physics. As the reviewer correctly pointed out, what we stressed is the asymmetry in resulting deformations. A more appropriate term might be power transmission. Because we did not perform any power analysis and because force or displacement seems more explicit for non-physics (physiology) readers, we chose the loose expression. Per reviewer's feedback, the title, abstract, and discussion were revised.

(10) tissue preparation Could you give the longitudinal extend of your excised OoC prep? How many longitudinal OHCs are in your preparation? Longitudinal interactions are also quite important for tuning and possibly for the distribution of motion energy coming from the OHCs.

This information on longitudinal span was provided in Fig. S2. Now the longitudinal span of our preparation is stated in the Methods section. Our preparation was prepared with the intention of minimizing the longitudinal interactions. By widely opening the cochlear tubes, acoustic pressures were delivered uniformly along the open span. As a result, the primary consequence of fluid-mediated longitudinal coupling, the traveling waves, virtually disappeared in our preparation. In other words, our preparation reduced 3D mechanics in to 2D mechanics so that we focus on the motion in a radial section.

(11) Stimulation: What were the voltage and the voltage steps you applied? Please try to give some assessment what this could imply for transmembrane voltage across OHCs.

We added a new paragraph in the Discussion. The transmembrane alternating current was typically 0.1 mA in amplitude, and the transmembrane resistance was typically 5 kΩ. Thus, our voltage amplitude was 500 mV. We estimate that the change in the transmembrane potential is equivalent to 3-10 mV in intact cochlea.

Reviewer #3 (Remarks to the Author):

The authors present a paper based on well conducted experiment on an in vitro preparation of the 4kHz frequency region of the gerbil cochlea using cutting edge techniques to address and attempt to resolve conflicting outcomes from recent papers using similar techniques on largely in vivo preparations. In general, the paper is well written, and the data are presented clearly. However, there is almost no

distinction made throughout the paper between in vivo and in vitro measurements, between measurements made at different frequency regions, and between measurements and theory. These distinctions are important because, in addition to its complex cellular architecture, which differs in a graded manner along the cochlea, the cochlea has a complex electrochemistry that strongly determines its electromechanical properties and performance. These properties are often neglected in theoretical papers and can be compromised in vitro studies. The majority of comments below are directed at these considerations.

A section was added to the Discussion to distinguish and compare the present study from in vivo studies. And a clearer distinction was provided in Methods as well.

Page 2 Abstract

Line 5. “recent data”. Please specify the type of data, the reviewer believes it to be in vivo data. It was revised to “recent data from live cochlear experiments”.

Line 7. Please replace “we present motion measurements” with “We present motion measurements from an isolated in vitro preparation”

Done

Page 3 Introduction

Line 7. Please replace “the hair cells” with “the hair cells as indicated by recent theoretical models”

Done

Lines 8-9 “Meanwhile, the OoC is flexible enough to get deformed by motile outer hair cells.” Is this based on measurements or models?

Line 9. “Meanwhile, the OoC is flexible enough to get deformed by motile outer hair cells.” Please support these two important statements above by references to direct measurements. If these are not available, then by references to theoretical studies.

Both. Proper citations were added. (Cooper, van der Heijden, 2018; Liu, et al. 2019; Zhou et al., 2022; Samars, et al., 2023).

Lines 34- 35 “First, in live cochlear experiments, it is tricky to isolate active OoC motion due to outer hair cell motility from passive motion due to acoustic pressures.” Please state the basis for this statement. For example, it is possible to target prestin without changing the passive mechanical properties of OHCs and to remove the influence of cochlear amplification and compression, thereby causing the alter mechanical and electrical responses of the cochlear to be linear over their entire Deiters’ level and frequency range (e.g. Dallos et al., <https://doi.org/10.1016/j.neuron.2008.02.028> , Weddell et al [10.1016/j.cub.2011.08.001](https://doi.org/10.1016/j.cub.2011.08.001)). It has however, been difficult both in vivo and in vitro to isolate Deiters’ cell activity from OHC motility without selectively targeting Deiters’ cells e.g. Zhao HB et al Neurophysiol. 2022 Jan 1;127(1):313-327. doi: 10.1152/jn.00468.2021. Lukashkina doi: 10.1523/JNEUROSCI.2127-21.2022

We see two points in this comment. First, the isolation of active OoC motion. The targeted removal of

prestin can isolate passive mechanics, but that method does not “isolate” active mechanics (motion purely due to outer hair cell motility, excluding the motion due to fluid pressures). The difference in cochlear responses between wild-type and prestin-deficient animals informs the loss in cochlear sensitivity, but not the underlying OoC micro-mechanics. Our study does not and cannot discuss cochlear sensitivity or amplification. Our study is regarding OoC micro-mechanics, which is a building block for active cochlear mechanics. Second, the Deiters cell. To our understanding, the referenced articles might be interpreted as Deiters’ cells can modulate outer hair cell motility. We agree that the view has merit as was studied in our previous theoretical and experimental studies (Nam, 2014; Zhou et al., 2022). As the references provided by the reviewer indicated, distinguishing the Deiters cells’ contribution requires careful considerations. Although that view of the Deiters cell as outer hair cell modulator is out of the scope of our present study, we wish to dedicate to the topic in the future.

Line 40. “relaxed”. Perhaps try “avoided”, or perhaps “addressed”?

Revised.

Lines 41-42. “By reducing the bones in the optical path, the vibrometry image “

Perhaps say something like: “By removing structures that would normally obstruct the optical path.....”

Revised.

Introduction page 2

Lines 3-4 “We focused on 2-D motions of the basilar membrane, reticular lamina, and outer pillar cell because they construct the frame for active and passive force transmission in the OoC.”

This is fine for studying radial and transfer motion, but not for longitudinal motion. Perhaps specify radial and transverse, because others have shown that longitudinal flow of energy plays important roles in shaping the active and passive mechanical properties of the OoC.

In this work, we simply stated that we focused on the motion in a radial section (i.e., transverse and radial motions, but not longitudinal motions). This does not mean that longitudinal motion is negligible. Note that, in existing in vivo studies, observing motion at an exact radial section is not straightforward. The uncertainty of the view plane causes ambiguity in the interpretation of data, as pointed out by Cooper et al. (2020), Frost et al. (2023), He, Burwood et al. (2022), and Meenderink and Dong (2023).

To make this point clearer, we revised the statement.

Lines 5-7. “Our results provide a coherent explanation of how outer hair cells exploit their structural frame to enhance vibrations of the upper part of the OoC while minimizing interference with the basilar membrane.”

This could be misleading and controversial as it is written. It might be more accurate to say “Our results provide a coherent explanation of how outer hair cells exploit their structural frame to enhance vibrations of the upper part of the OoC relative to those of the basilar membrane.”

We revised.

Results

Page 3 lines 6-8. Why didn’t you include Deiters’ cells in your study? There are in vivo and in vivo studies that show their activity influences OHC motility, they can control cochlear sensitivity, and have been identified as the essential elements of feedforward models of longitudinal force transfer in the OoC. e.g. Yoon et al., Biophysical Journal 100:1-10.

This study is dedicated to ‘presumably’ stiff structures that frame the outer hair cells. This being said, we strongly agree with the reviewer about the importance of the Deiters cells. We reported our studies

dedicated to the Deiters cell previously (Nam 2014; Zhou et al., 2022). We admit they were far from sufficient. For the active contribution of Deiters cells, as the reviewer notes above, it must take cautious considerations to better reflect physiological conditions. We certainly are aiming to investigate in that direction.

Lines 11-12. “of passive (dead or insensitive) and active (sensitive) cochlear responses of the intact cochlea, respectively (Fig. 2).” What are the criteria for categorizing “active” and “dead”. There appears to be nothing about this in the methods. It is important that the reader understands the criteria for this distinction.

Thanks for this point that we overlooked. We defined the term ‘active’ more clearly at the beginning of Results.

Lines 13-14. “In contrast, the vibrating pattern was non-monotonic for E stim (Fig. 2H and J).” This conclusion involves some unknowns over which you have no control. These include: Mechanical stimulation may or may not cause OHC electromotility depending on the state of the preparation and the stimulus frequency relative to the frequency of the measurement location. Electrical stimulation could drive both OHC electromotility and Deiters cell electromotility, as demonstrated directly in vitro (Yu, Zhou PLoS ONE 4: e7923) depending on the state of the preparation. This latter possibility could cause changes to the OHC structural cage that are not encountered in vivo. Can you please comment on this because it is important for understanding and interpreting your findings. One way out of this apparent dilemma is to suggest that during electrical stimulation, Deiters’ cell motility may always accompany OHC motility (e.g. Lukashkina et al, Journal of Neuroscience 42:5660-71), and modify OHC motility (e.g. Yu, Zhou PLoS ONE 4: e7923). With mechanical stimulation, the mechanics and the electrochemistry of the preparation has been altered so that it resembles a passive cochlea.

There are several heavy points in this comment.

There was little to no electromotility due to M-stim. First, the endocochlear potential was abolished. Second, the motion amplitude by M-stim was typically on the order of 100 nm which is much greater than expected residual OHC motility (if there by any). Indeed, the M-stim responses were strictly linear indicative of little OHC feedback. Finally, we tested with hair cell MET channel blockers and the responses were indistinguishable. The challenge was the opposite way around (i.e., we wanted to make sure there was no sign of MET current).

The focus of this study is at the strong neighbors of the outer hair cell: the pillar cell, reticular lamina and basilar membrane. Although we did not analyze the motions of the Deiters cell, we agree that it is a great topic for our next study. Thus far, we have not seen any sign of Deiters cell electro motility. Had we observed so, it must be a big point to report. In the reviewer’s referenced papers, there was no strong evidence of Deiters cell electromotility. The evidence rather suggests that the Deiters cell may contribute to setting the operating point of OHCs.

Lines 18-19. “The motion trajectories were transversely dominant in M-stim responses, while E-stim responses showed greater radial motions, especially around outer hair cells.”

Radial vibration of the BM may vary with species and frequency place on the BM and it is thought this point should be made clear. It would be interesting to compare these parameters in the 70 kHz region of the mouse cochlea with the 4 kHz region in the gerbil.

We agree. We were careful not to generalize that our observations applied universally across different frequency gamut and species.

Page 4. Lines 1-14. This section deals with several major issues that the reviewer believes should be addressed to make your findings more useful in understanding the electromechanical properties of the cochlear partition.

The first of these is that the structure of the cochlear partition. There are many references available describing the structure of the cellular components and of the structural frame encasing the sensory electromotile outer hair cells, and indeed the OHCs themselves, that show strong structural differences from apex to base. These differences are reflected in the in vivo physiology and has led to the suggestion that apical regions of the cochlea process isotonic forces, while the basal region processes isometric force (<https://doi.org/10.1038/s41598-017-04279-3>). Thus, the transvers and radial displacements that you very elegantly demonstrate in your paper, may not be replicated in more basal regions and it is recommended that you stated this, or something to that effect, because it avoids criticism and paves the way for further measurements to address this question.

The second concerns the interpretation of the outcomes from Mechanical and Electrical stimulation of the preparation (see Methods). The outcomes from both forms of stimulation depend on the physiological state of the preparations, which might not be ideal considering the mechanical trauma, the time it takes to make the preparation before measurement, and the substitution of endolymph for perilymph. The sensitivity of in vivo preparations depends critically on the sensitivity of the cochlea. A difference in sensitivity of 5dB between preparations can alter the mechanical and electrical properties of the cochlea, for example in terms of saturation and nonlinearity and in the mechanical responses of the cochlear to acoustic stimulation, which may also account for differences in the reported mode of vibration across the width of the cochlea (P4,L38-39). Thus, in response to mechanical stimulation, the preparation may be that of a passive cochlea, without amplification. Electrical stimulation may cause the OHCs to change length, as appears in Figs. 4-6. In addition, it Deiters' cells, at least, are motile (e.g. Dulon et al, 1994 doi: 10.1006/bbrc.1994.1841) and could contribute to the complex motion of the cochlear partition during electrical stimulation, whereas they may contribute indirectly by influencing OHC motility during low-level mechanical stimulation in vivo (Lukashkina et al., 2022). Thus, the responses of the structural framework surrounding the OHCs to electrical stimulation may resemble those measured in vivo in sensitive cochlea to mechanical stimulation, but perhaps with contributions from elements, such as the Deiters' cells, which would not normally contribute to the movement of the structural framework during near threshold mechanical stimulation. It is recommended that the authors revise this section and take into account the points raised above.

Hopefully, the newly added section in Discussion addressed this point. As the reviewer said, because cochlear sensitivity is highly fragile, keeping the cochlea sensitive takes careful attention even in live animal experiments. An excised cochlear preparation cannot be sensitive. Our preparation is definitely passive. Meanwhile, outer hair cells can remain motile even within an insensitive cochlea for several hours. Our study is regarding the OoC vibrating patterns due to outer hair cell motility. As we responded in the above, the Deiters cell mechanics is out of the scope of this study. With our current data, we are not ready to discuss its role in 'active' force transmission (we discussed its passive force transmission in

our previous paper, Zhou et al., 2022). We definitely share the reviewer's interest in the Deiters cell. The Deiters cell is greatly positioned to modulate the action of the cochlear actuator (outer hair cells).

With regards,

Jong-Hoon Nam

Reviewers' comments:

Reviewer #1 (Remarks to the Author):

I have now read the authors responses to the reviewer's comments, and the revised manuscript, and I would like to thank the authors for some clarifications regarding the comments I had.

The new paragraph which was added to the discussion regarding the sensitivity of the preparation is certainly helpful, and the definitions of what the authors call "active" and "passive" responses might limit the confusion the use of these terms might cause in this context, especially for auditory physiologists working in-situ.

I understand now better that the aim of this paper is to dissect Organ of Corti micromechanics in response to mechanical ("passive") and electrical ("active") stimulation, and the authors claim to be able to clearly separate the two. They, not unreasonably, assume that there is very little outer hair cell mechano-electrical transduction due to the nature of their preparation, which should result in a presumably exclusively passive response (i.e. without OHC electromotility) to mechanical stimulation. The authors state in a response to another reviewer's comment that they used MET channel blockers to prove this point, and found no difference compared to measurements without channel blockers. I would like to suggest including the data of this important control experiment (or refer to one of their previous papers if they included such a control experiment), as the authors stress on various occasions the clear separation between passive, and active, outer hair cell-driven movements as an important feature of their preparation.

While the focus of this paper is to characterize force transmission in the Organ of Corti, not resembling a physiological situation, our aim must be to better understand the Organ of Corti mechanics in a physiological situation. I was therefore wondering whether the OHC electromotility observed with electrical stimulation in this preparation is likely to resemble electromotility in an uncompromised cochlea? In this context am I also a little surprised that the authors state that it was not possible to drive the electro-mechanical transduction into saturation, as the operating point of the electro-mechanical transduction might be quite shifted compared to physiological situations, due to the most likely altered OHC resting membrane potential.

Reviewer #2 (Remarks to the Author):

The authors changed their manuscript according my suggestions in a satisfactory way. I have no further comments and the ms could be published as it is.

Reviewer #3 (Remarks to the Author):

Reviewer 3 has read the responses made by the authors to the comments made by all reviewers. The authors appear to have taken the comments very seriously and made strong attempts to address them.

The view of reviewer 3 is that, in spite of the addition to the discussion and changes to the text to address the comments by the reviewers, there are still shortcomings in the paper that make interpretation of the findings open to question.

See specific question 1, reviewer 1.

For example, there is published evidence (Brundin, Russell, Hear. Res.1994 73:35-45) that isolated OHCs quickly lose their turgor pressure. Without controlling for this factor in the isolated preparation, the mechanical properties of the OoC could be changed, thereby influencing the reported findings.

Specific question 4, reviewer 1 regarding Salicylate. These points and earlier ones are covered in the

findings and discussion in Drexl et al., 2008 (J Neurophysiol 99: 1607–1615, 2008). In summary, Salicylate reduces prestin-mediated electromotility, reduces the turgidity of OHCs and longitudinal stiffness. Salicylate does not completely abolish electrically elicited responses, but through use of different mouse mutants, the residual electrically elicited motility is most likely a product of electrokinetic forces due to the movements of charged membranes and not to residual OHC electromotility. These earlier findings by Drexl et al could account, in whole or in part, for the complex movements of the cochlear partition measured by the authors during electrical stimulation.

Reviewer 2 response to specific question 2(2) results, page 3, line 11

See above. The lack of control of the state of the OHCs and their role in the mechanics of the cochlea is an important omission. For example, loss of turgor pressure and loss of prestin influences the stiffness of the cochlear partition (see Drexl above and Mellado Lagarde (Curr Biol. 2008 Feb 12;18(3):200-2.)

Reviewer 2 question 7: (7)

This raises an important point that was not made clear in the paper. In most mammals, but not in some bat species, the structure of the OoC changes with location being very compliant, with very flexible inner pillar and Deiters' cells (which may even be absent) at the apex and far stiffer structures at the base of the cochlea as the forces generated by the OHCs are suggested to change from isotonic to isometric (see: Lukashkina et al, Sci Rep. 2017 Jul 12;7(1):5185.). In the high frequency region of the bat cochlea, the Deiters' cells are massive compared to the OHCs, and equipped with a special type of tubulin.

Reviewer 2 question 9

The symmetry and asymmetry may be a consequence of several factors, including the mechanical properties of the region from which the measurements are made. A point not made clearly apparent in the paper. It is widely understood that OHCs react against the very much stiffer BM and radial components of this force lead to the excitation of the OHCs. In the stiff, basal turn of the cochlea, the OHC receptor potential is symmetrical. This is not the case for the asymmetry of OHC receptor potentials measured in the more compliant apical turns of the cochlea (e.g. Nature volume 429, pages 766–770 (2004)).

The authors appreciate the thoughtful critiques of the reviewers. Below we respond to reviewer's comments point-by-point.

Reviewers' comments:

Reviewer #1 (Remarks to the Author):

I have now read the authors responses to the reviewer's comments, and the revised manuscript, and I would like to thank the authors for some clarifications regarding the comments I had.

The new paragraph which was added to the discussion regarding the sensitivity of the preparation is certainly helpful, and the definitions of what the authors call "active" and "passive" responses might limit the confusion the use of these terms might cause in this context, especially for auditory physiologists working in-situ.

I understand now better that the aim of this paper is to dissect Organ of Corti micromechanics in response to mechanical ("passive") and electrical ("active") stimulation, and the authors claim to be able to clearly separate the two. They, not unreasonably, assume that there is very little outer hair cell mechano-electrical transduction due to the nature of their preparation, which should result in a presumably exclusively passive response (i.e. without OHC electromotility) to mechanical stimulation. The authors state in a response to another reviewer's comment that they used MET channel blockers to prove this point, and found no difference compared to measurements without channel blockers. I would like to suggest including the data of this important control experiment (or refer to one of their previous papers if they included such a control experiment), as the authors stress on various occasions the clear separation between passive, and active, outer hair cell-driven movements as an important feature of their preparation.

In response to the reviewer's suggestion, we re-performed a MET channel blocker experiment (Fig. R1). We measured M-stim responses with and without MET channel blockers (0.1 mM curare) for three pairs of application-washout rounds. We chose curare because its effect time is known to be faster than other well-known MET blockers (Glowatzki, E., et al. 1997). The displacements at the reticular lamina and basilar membrane were both indistinguishable in the control and test groups. We prefer not to include this result in the text because this does not add information to our main text.

While the focus of this paper is to characterize force transmission in the Organ of Corti, not resembling a physiological situation, our aim must be to better understand the Organ of Corti mechanics in a physiological situation. I was therefore wondering whether the OHC electromotility observed with electrical stimulation in this preparation is likely to resemble electromotility in an uncompromised cochlea? In this context am I also a little surprised that the authors state that it was not possible to drive the electro-mechanical transduction into saturation, as the operating point of the electro-mechanical transduction might be quite shifted compared to physiological situations, due to the most likely altered OHC resting membrane potential.

In our preparation, OHC electromotility due to transmembrane potential change was driven by alternating current source (E-stim) but not MET current. As a result, transmembrane potential change was not controlled by the operating point of the electro-mechanical transduction but by the level of applied currents. In our previous response, we stated that it was not feasible to drive OHC motility (not MET) into saturation. Nonlinearity due to MET saturation is not the scope of this study.

This study regards OoC mechanics in response to acoustic pressure and OHC force. As long as OoC structures remain undamaged (new Fig. S4), we argue that OoC mechanics are preserved. Whether OHC motility is driven by MET current or extracellular current, as long as the motile OHC deforms the OoC, our conclusions remain the same. In our preparation, a physiological condition (MET) was compromised with purpose, but physics (OoC micro-mechanics) was close to intact condition. To further support this latter point, we newly present Fig. S4.

Reviewer #2 (Remarks to the Author):

The authors changed their manuscript according my suggestions in a satisfactory way. I have no further comments and the MS could be published as it is.

Thank you.

Reviewer #3 (Remarks to the Author):

Reviewer 3 has read the responses made by the authors to the comments made by all reviewers. The authors appear to have taken the comments very seriously and made strong attempts to address them.

The view of reviewer 3 is that, in spite of the addition to the discussion and changes to the text to address the comments by the reviewers, there are still shortcomings in the paper that make interpretation of the findings open to question.

(1) See specific question 1, reviewer 1.

For example, there is published evidence (Brundin, Russell, Hear. Res.1994 73:35-45) that isolated OHCs quickly lose their turgor pressure. Without controlling for this factor in the isolated preparation, the mechanical properties of the OoC could be changed, thereby influencing the reported findings.

This is a fair point. We considered this matter of OHC integrity. Previous studies assessed OHC turgor pressure using morphometry (Brownell & Shehata, 1990; Brundin & Russell, 1994; Chertoff, Brownell, 1994; Chan, Ulfendahl, 1997; Belyantseva et al., 2000; Kakethata, Santos-Sacchi, 1995). Likewise, we assured the tissue (OHC) integrity by monitoring the morphometric changes of OHCs. Unlike most previous studies regarding OHC turgor pressure, our OHCs are *in situ* (not isolated from their framing structures).

To verify our prepared tissue, including the OHCs, was in normal morphometric condition, a figure (Fig. S4) was newly added in the Supporting Materials. For the convenience of the reviewers, Fig. S4 is appended in this response as well.

To our knowledge, no one has measured OHC turgor pressure explicitly either *in situ* or *in vivo*. OHC turgor pressure was estimated from the geometry or texture of OHCs. Unlike the cases of *ex situ* OHCs, we did not see any evidence of OHC turgor pressure change from our OHCs *in situ* over a time span of 2-3 hours from animal death (Fig. S4A, B, D, E). Even though we observed modest morphological changes after 3 hours postmortem (thicker OHCs, Fig. S4C), we could not tell differences in OoC mechanics. Despite this, most of our presented measurements were made within 2.5 hours of animal death. When we noticed deformed cells, the experimental data were excluded (dramatic examples in Fig. S4G-J).

With the fine optical resolution of our OCT system, we could detect and record OHCs' morphological changes. Fig. S4 presents normal (included) versus abnormal (excluded) preparations. Three types of abnormal morphology were illustrated: swollen OHCs (Fig. S4G), detached tectorial membrane (Fig. S4H), and lost Deiters' cells (Fig. S4I, J). Please, do not get us wrong: poor tissue integrity was observed rarely. Deformed OHCs in the early stage of an experiment (< 2 hours) have not been observed since April 2022. The lost Deiters' cells in Fig. S4I and J were an accidental case, observed in 2021, potentially due to a wrong solution. On the other hand, the tectorial membrane was trickier to keep its intact shape. Over the span of the present study (June 2022 - January 2023), partly or fully detached tectorial membranes were observed in 4 out of 26 measurements.

(2) Specific question 4, reviewer 1 regarding Salicylate. These points and earlier ones are covered in the findings and discussion in Drexel et al., 2008 (J Neurophysiol 99: 1607–1615, 2008). In summary, Salicylate reduces prestin-mediated electromotility, reduces the turgidity of OHCs and longitudinal stiffness. Salicylate does not completely abolish electrically elicited responses, but through use of different mouse mutants, the residual electrically elicited motility is most likely a product of electrokinetic forces due to the movements of charged membranes and not to residual OHC electromotility. These earlier findings by Drexel et al could account, in whole or in part, for the complex movements of the cochlear partition measured by the authors during electrical stimulation.

It is good to see this comment congruent with our argument that applying salicylate cannot totally separate passive and active mechanics. We strongly believe the movements of OoC due to E-stim could not be ascribed to other cells than OHCs. The supporting cells' deformation during E-stim is consistent with passive deformation caused by OHC motility (Zhou et al., 2022, this study). We believe that, even if there exists electrokinetic motion of other supporting cells, that must be negligible as compared to OHC motility in our preparation. We will consider future investigations to examine the findings of Drexel et al. more carefully.

(3) Reviewer 2 response to specific question 2(2) results, page 3, line 11

See above. The lack of control of the state of the OHCs and their role in the mechanics of the cochlea is an important omission. For example, loss of turgor pressure and loss of prestin influences the stiffness of the cochlear partition (see Drexel above and Mellado Lagarde (Curr Biol. 2008 Feb 12;18(3):200-2.)

We cannot see how the change in OHC stiffness can affect cochlear macro mechanics, especially in the basal region of the cochlea. According to explicit stiffness measurement by Naidu and Mountain (2001), the mechanical contribution of the OoC to overall stiffness is negligible compared to the basilar membrane, and most mechanical contribution of OoC came from the pillar cells not from the OHCs. We have studied this similar topic extensively: Even a tenfold reduction of the OHC stiffness in the middle turn of the gerbil cochlea will affect the macro mechanics minimally (Liu et al., 2015; Liu et al., 2017; Zhou et al., 2022). A more plausible explanation of the prestin KO studies is that missing prestin affected the development of other OoC structures as well as the OHCs. OHCs cannot have the level of turgor pressures that can substantially affect 'passive' mechanics. With this said, we agree that the change in

OHC turgor pressure can shift the operating states of OHC motility (Kekahata, Santos-Sacchi, 1995). Our study does not cover such nonlinear conditions.

Reviewer 2 question 7: (7)

This raises an important point that was not made clear in the paper. In most mammals, but not in some bat species, the structure of the OoC changes with location being very compliant, with very flexible inner pillar and Deiters' cells (which may even be absent) at the apex and far stiffer structures at the base of the cochlea as the forces generated by the OHCs are suggested to change from isotonic to isometric (see: Lukashkina et al, Sci Rep. 2017 Jul 12;7(1):5185.). In the high frequency region of the bat cochlea, the Deiters' cells are massive compared to the OHCs, and equipped with a special type of tubulin.

This is a very interesting point and indeed relevant to this study. We believe that Lukashkina et al.'s study is in line with our argument. The mammalian cochlea needs a mechanism to accommodate such an extreme range from isotonic to isometric. Our results suggest a mechanism for OHC force being delivered to the top side of the OoC even when the structure is compliant (in the apical turn of the cochlea). Meantime, this paper is dedicated to OoC micromechanics (not macro-mechanics such as traveling waves, compressive nonlinearity, and tuning) based on data from a specific location of a specific animal model (the middle turn of the gerbil cochlea). Exploring location- and species-dependent OoC micromechanics will certainly be our future effort. We read our MS again to make sure that we do not overstate our work as if our results apply universally despite locations and species.

Reviewer 2 question 9

The symmetry and asymmetry may be a consequence of several factors, including the mechanical properties of the region from which the measurements are made. A point not made clearly apparent in the paper. It is widely understood that OHCs react against the very much stiffer BM and radial components of this force lead to the excitation of the OHCs. In the stiff, basal turn of the cochlea, the OHC receptor potential is symmetrical. This is not the case for the asymmetry of OHC receptor potentials measured in the more compliant apical turns of the cochlea (e.g. Nature volume 429, pages 766–770 (2004)).

We share the reviewer's interest regarding compliant OHCs working on stiffer BM. Our mentioned asymmetry is a different type of asymmetry from what the reviewer says. Considering the absence of MET in our preparation, the asymmetry due to OHC receptor potential cannot be investigated in this study. Our asymmetry is limited to OoC micromechanics during phasic force transmission.

Figure S4. The structural integrity of experimental preparations. B-scan images are from four sample cochleae. The B-scan images inform the structural integrity of our preparations. (A-C) B-scan images of Fig. 6 preparation at the time points from animal death (indicated in the top right corner of each panel). (D-F) B-scan images of Fig. 4 preparation at three time points. (G-I) Three examples of data excluded due to abnormal morphology: swollen outer hair cells (G), detached tectorial membrane (H), and lost Deiters' cells (I). (J) longitudinal scan of the same cochlea as (I). The green broken lines indicate where the corresponding radial/longitudinal sections were taken. These negative examples were acquired while refining current experimental protocols before obtaining the data set used for this study. The data in this paper were obtained between June 2022 and January 2023. The last observation of badly swollen outer hair cells was in April 2022. Lost Deiters' cells are rare (they have not been observed since 2021). Meanwhile, delaminated tectorial membranes were observed during the experimental period of this study. Detached TM was observed in 4 out of 26 measurements. Scale bars: 100 μm . Data sets used: 0822, 0826, 0304 of year 2022, 0919 of year 2021.

REVIEWERS' COMMENTS:

Reviewer #1 (Remarks to the Author):

I have now read the responses to my questions and comments and would like to thank the authors for their time. I have no further comments and feel that the manuscript is now ready for publication.

Reviewer #3 (Remarks to the Author):

I thank the authors for their considered and thorough responses to my questions and I am satisfied with their responses.